# CLOE: CHRISTOFFEL LOss AUTOENCODER FOR ANOMALY DETECTION

## ABSTRACT

Semi-supervised anomaly detection plays a key role in diverse fields such as process monitoring, healthcare, and finance. However, lightweight methods often struggle with high-dimensional data and typically require careful tuning of multiple hyperparameters. Among existing approaches, Christoffel Function–based methods are attractive due to their simplicity, requiring at most a single hyperparameter. They also benefit from a well-established theoretical foundation that yields several interesting results for data science. Their main limitation, however, is poor scalability to high-dimensional settings. In this paper, we introduce CLOE, a new method that combines an autoencoder for dimensionality reduction with a Christoffel Function–based detector applied in the latent space. To better align representation learning with anomaly detection, we design a novel loss function that leverages the Christoffel Function to guide the autoencoder toward representations that better capture the support of the normal data distribution. We further propose a principled procedure to set the detection threshold and an efficient strategy to tune the single remaining hyperparameter. Experiments on multiple high-dimensional anomaly detection benchmarks demonstrate that CLOE achieves superior performance compared to existing methods, while preserving the lightweight and low-tuning advantages of Christoffel Function–based approaches.

## 1 INTRODUCTION

The growth in sensor deployment for monitoring activities in health, industry, and other domains is creating substantial amounts of high-dimensional data. A crucial application is anomaly detection (AD), i.e., identifying abnormal or rare events, known as outliers. In semi-supervised learning, AD methods are trained using samples known to be normal (inliers). These methods estimate the distribution of the data and compute a score for each test sample. To detect outliers, the score is compared to a threshold provided by the method (Platt et al., 2001). However, most classical AD methods are challenged by the curse of dimensionality and do not consider the full complexity of data. The time complexity to estimate a distribution is very high, not always suitable for non linear settings, and cross-variable dependencies are not taken into account (Samariya & Thakkar, 2023) and (Pang et al., 2021). Among the various methods, those based on the Christoffel Function (CF) have drawn our attention (Ducharlet et al., 2024). Rooted in approximation theory and orthogonal polynomials, the CF is grounded in a rigorous algebraic framework that addresses key requirements of data science (Lasserre et al., 2022), particularly the need to be free from hyperparameter tuning (Ducharlet et al., 2024).

Deep learning offers a solution to handle high-dimensional data. A neural network can reduce the dimensionality of the data while considering cross-variable dependencies. Autoencoders (AE), a class of neural networks, consist of an encoder and a decoder that are trained to reconstruct the input data while reducing the data dimensionality in the latent space in a nonlinear way (Wang et al., 2016). The encoder hence encodes data in a low-dimensional space so that a classical AD method can be used to detect outliers using the latent space. However, the learned representations may not optimally capture the support of the normal data for anomaly detection. To address this, the training of the autoencoder can be guided by the anomaly detection method, ensuring that the latent space provides more informative and discriminative representations. This principle is known as coupled or joint training (Huang et al., 2025).

In this paper, we propose CLOE (Christoffel LOss for autoEncoder), an efficient approach for high-dimensional tabular anomaly detection in a one-class classification setting, i.e., only normal samples are available during training. In CLOE, an AE reduces data dimensionality, and its latent space is regularized using the empirical Christoffel Function (CF) (Lasserre & Pauwels, 2019), a concept from approximation theory. By introducing CF-based loss, that is differentiable, during training, CLOE learns representations tailored for defining compact normal data supports, enabling robust outlier detection by the subsequent CF-based anomaly detection method applied to the latent space. Moreover, a particular advantage of the CF method is that it only requires one hyperparameter to be set. CLOE is computationally lightweight and designed to operate on CPUs, which is well-suited for resource-constrained environments. This method has been developed in an industry context and will be trained in a lot of different high-dimensional datasets. It requires the less computational resources possible to be trained and inferred.

The main contributions of this paper are summarized as follows:

- We introduce the new method CLOE, which performs effective representation learning in a lower-dimensional latent space guided by the empirical CF for tabular data, using a lightweight computational approach that does not require GPU acceleration;

- We propose a process for selecting the single hyperparameter of the model, eliminating the need for extensive hyperparameter tuning;

- We conduct comprehensive experiments on 15 high-dimensional tabular datasets from the ADBench benchmark.

## 2  RELATED WORK

AD methods can be classified into two different types: classical and the deep learning AD methods.

A classic way to detect outliers in a cloud of points is to estimate density, like Density-Based Spatial Clustering of Applications with Noise (DBSCAN) (Ester et al., 1996), Kernel Density Estimation (KDE) (Parzen, 1962), Histogram-Based Outlier Score (HBOS) (Goldstein & Dengel, 2012), or Empirical Cumulative Distribution for Outlier Detection (ECOD) (Li et al., 2022). After density estimation, data points within low density regions are considered outliers. Another approach is to compute the distribution support used to define the boundary of normal data like One-Class Support Vector Machine (OC-SVM) (Schölkopf et al., 1999), Support Vector Data Description (SVDD) (Tax & Duin, 2004), and the empirical CF (Lasserre & Pauwels, 2019). After support computation, data points lying outside the support are then considered as outliers. A simpler method can be to compute the distance between the k-nearest neighbors (kNN) (Ramaswamy et al., 2000) of each sample and consider those with largest distances as outliers. However, these classical AD methods do not scale well with high-dimensional data. For example, the computation time can become prohibitively high for the empirical CF (Ducharlet et al., 2024), or interdependencies between dimensions are lost in HBOS and ECOD (Han et al., 2022).

To address these challenges, deep neural network (DNN) AD methods have been developed. Most of these approaches are semi-supervised, trained only on normal samples. The DNN is trained to reconstruct the input sample and the outlier score is computed as the difference between the input and the reconstructed output. RCA (Liu et al., 2021) considers many AEs and uses the $k$ samples with the lowest reconstructed scores of an AE to train the other AEs. MCM (Yin et al., 2024) trains a generator to mask inputs and trains an AE to reconstruct the masked inputs. These methods can be more complex and train a neural network to reduce data dimensionality and then feed reduced data into a classical AD method to identify the outliers. DeepSVDD (Ruff et al., 2018) extends the SVDD method by learning useful data representations and optimizing the SVDD objective. Latent Anomaly Detection through Density Matrices (LADDM) (Gallego-Mejia et al., 2024) builds a density matrix with the encoded data transformed into a Hilbert space. Adaptations of Deep-Clustering (DEC) (Xie et al., 2016) have led to deep clustering-based anomaly detection methods: the AE is first pretrained with the reconstruction error, and training then continues with a clustering-based loss. DEC proposes a k-means-based loss (Xie et al., 2016) while Deep-Clustering Compact (DCC) (Arellano-Espitia et al., 2021) utilizes an OC-SVM-based loss. These methods construct new representations of the data points and then fed into a classical AD method. However, these newly

learned representations may lose information relevant for AD, making the classical AD method less effective (Pang et al., 2021).

A solution is joint training, where the autoencoder is trained with a loss that combines the reconstruction error and a loss term from the downstream classical AD method. This approach guides representation learning and improves AD performance. The Deep Clustering Hierarchical AutoEncoder (DCVAE) and Deep Nested Clustering AutoEncoder (DNCAE) (Nguyen et al., 2024) extend deep-clustering methods by using either a double autoencoder or different layers of the same autoencoder to produce multiple representations of the data. These representations are used to compute a k-means clustering-based loss summed with the reconstruction error. The Deep Autoencoder Gaussian Mixture Model (DAGMM) (Zong et al., 2018) combines the reconstruction error of the autoencoder with the latent space representation to feed a neural network that outputs the mixture membership predictions for each data point. The parameters of the GMM are then estimated, and each sample's energy is computed. The model is jointly trained by optimizing the reconstruction error and the sample energy. OCSVM-Guided representation learning (Og) (Pinon & Lartizien, 2025) trains an autoencoder with a loss that combines the reconstruction error and an OC-SVM-based loss. However, such losses are not always differentiable, as in Og, so training the model with backpropagation can assign arbitrary gradient values at the non-differentiable points of the losses (Paszke et al., 2019). Finally, Decomposed Representation Learning (DRL) (Ye et al., 2025) proposes a low-dimensional data representation where the representations of each normal sample are decomposed into a weighted linear combination of randomly generated orthogonal basis vectors.

The central idea of this paper is to use a CF-based method as the downstream AD method because it offers theoretical proofs for support estimation and outlier detection. However, this method does not scale to high dimensional data. A deep neural network is used to reduce high-dimensional data, with a joint training guided by the CF, to propose data representations adjusted for the CF-based anomaly detection.

## 3 BACKGROUND

The CF is a well-known concept in approximation theory. Recent studies (Lasserre & Pauwels, 2019) and, (Lasserre et al., 2022) propose to adapt it to data analysis as a means to estimate the support of a distribution, which may be highly nonlinear. This section resumes some important definitions about the CF and its empirical counterpart from Lasserre & Pauwels (2019) and Lasserre et al. (2022).

### 3.1 PRESENTATION OF THE CHRISTOFFEL FUNCTION

Let $\Omega \subset \mathbb{R}^d$ be a compact set with non-empty interior. Let $\mu$ be a finite Borel measure supported on $\Omega$. $\mu$ is absolutely continuous w.r.t. Lebesgue measure on $\Omega$, a set with non-empty interior and positive density. Let $\boldsymbol{v}_n(x) := (P^\alpha)_{\alpha \in \mathbb{N}^d}$ be the monomial basis of the vector space of $\mathbb{R}[x]$ of all the monomials of degree less than or equal to $n$ graded in the lexicographic order[1]. The size of the vector $\boldsymbol{v}_n(x)$, denoted as $s_d(n)$, is equal to $\binom{d+n}{n}$.

**Definition 3.1 (The Christoffel Function)** *The Christoffel Function (CF) of degree $n \in \mathbb{N}$ associated with the measure $\mu$, denoted by $\Lambda_n^\mu(x)$, is defined as*

$$\Lambda_n^\mu(x) = \min_{P \in \mathbb{R}_n[x]} \left\{ \int_\Omega P^2(z) \, d\mu(z), P(x) = 1 \right\} \tag{1}$$

Let $\boldsymbol{M}_n(\mu)$ be the moment matrix of $\Omega$. $\boldsymbol{M}_n(\mu)$ is a real symmetric matrix, $M_n(\mu)$ can be written as

$$\boldsymbol{M}_n(\mu) = \int_{\mathbb{R}^d} \boldsymbol{v}_n(x) \boldsymbol{v}_n(x)^T d\mu(x) \tag{2}$$

$\boldsymbol{M}_n(\mu)$ is positive definite and is non-singular for all $n$ .

---

[1]lexicographic order: monomial are first sorted by degree and then using lexicographic order on variables considering $X_1 = a$, $X_2 = b$, etc.

For our study, we will consider the inverse of the CF. Let us introduce the Christoffel-Darboux kernel $K_n^\mu$ associated with $\mu$. Given any basis of $\mathbb{R}_N[\boldsymbol{x}]$, orthonormal with respect to the inner product induced by $\boldsymbol{M}_n(\mu)$, $(p_i)_{i=1}^{s_d(n)}$, $K_n^\mu$ is defined as:

$$(x, y) \mapsto K_n^\mu(x, y) := \sum_{i=1}^{s_d(n)} p_i(x) p_i(y). \tag{3}$$

This kernel can also be computed from the moment matrix:

$$(x, y) \mapsto K_n^\mu(x, y) := \boldsymbol{v}_n(x)^T \boldsymbol{M}_n(\mu)^{-1} \boldsymbol{v}_n(y). \tag{4}$$

Let the polynomial $Q_{\mu,n}$ be defined by

$$Q_{\mu,n}(x) = K_n^\mu(x, x) = \boldsymbol{v}_n(x)^T \boldsymbol{M}_n(\mu)^{-1} \boldsymbol{v}_n(x), x \in \mathbb{R}^d. \tag{5}$$

$Q_{\mu,n}$ is a sum-of-squares polynomial of degree $2n$, it is differentiable on $\mathbb{R}^d$. Pauwels & Lasserre (2016) showed that $Q_{\mu,n}$ has higher value for data points which are isolated from the other points. Lemma 4.3.1 (Lasserre et al., 2022) quantifies the exponential growth with $n$ for data points outside the support. Lemma 4.3.2 (Lasserre et al., 2022) quantifies the polynomial growth with $n$ for data points inside the support. The inverse of the CF is

$$\Lambda_n^\mu(x)^{-1} := Q_{\mu,n}(x), \forall x \in \mathbb{R}^d. \tag{6}$$

### 3.2 THE EMPIRICAL CHRISTOFFEL FUNCTION FOR DATA ANALYSIS

Let $\mathbb{X} \subset \mathbb{R}^D$ be a finite set of data of size $N$, $D > d$. Let $\mathbb{X}_e \subset \mathbb{R}^d$ be the encoded version of $\mathbb{X}$ in the $d$ dimension space. We consider the discrete measure $\mu_N$ whose support is $\mathbb{X}_e$ sampled from a theoretical measure $\mu$ supported on $\Omega$. The empirical version of the moment matrix can be written as

$$\boldsymbol{M}_n(\mu_N) = \frac{1}{N} \sum_{z \in \mathbb{X}_e} \boldsymbol{v}_n(z) \boldsymbol{v}_n(z)^T. \tag{7}$$

To guarantee the invertibility of the matrix $M_n(\mu_N)$, the size of $\mathbb{X}_e$ must be greater than $s_d(n)$ according to Lasserre et al. (2022), Corollary 6.3.5. Under the condition $|\mathbb{X}_e| = N > s_d(n)$, the inverse of the empirical CF is defined as:

$$\Lambda_n^{\mu_N}(z)^{-1} := \boldsymbol{v}_n(z)^T \boldsymbol{M}_n(\mu_N)^{-1} \boldsymbol{v}_n(z), z \in \mathbb{X}_e. \tag{8}$$

### 3.3 THRESHOLDING WITH THE EMPIRICAL CHRISTOFFEL FUNCTION

Outlier detection via the CF requires a thresholding policy. The CF is known to have theoretical properties in the analysis of discrete data to define level sets that capture quite accurately the geometric shape of the support (Lasserre et al., 2022). Lasserre et al. (2022) consider a problem in $\mathbb{R}^2$ in Chapter 7 and propose to fix the constant $n_N$ related to this problem introduced by Vu et al. (2022) to $n_N := \lfloor 2N^{1/4} \rfloor$. Then the empirical CF is evaluated at each point and the smallest value is chosen as threshold. This smallest value corresponds to the closest level set of the support of the normal distribution.

Ducharlet et al. (2024) propose a method, named DyCF, to detect outliers in data streams. The approach uses the Sherman-Morrison formula (Sherman & Morrison, 1950) to update the moment matrix for each new data point, avoiding the need to recompute its inverse at every step. The DyCF method requires only a single hyperparameter: the polynomial degree $n$. A scoring function is then defined as:

$$S_{n,d}(x) = \frac{\Lambda_n^{\mu_N}(x)^{-1}}{\gamma_{n,d}}, \tag{9}$$

where $\gamma_{n,d} = Cn^{3d/2}$. A point $x$ is detected as an outlier if $S_{n,d} \geq 1$.

A second method, named DyCG, proposes a solution free of hyperparameter tuning, that leverages the growth property of the CF. In DyCG, the scoring function is derived from the DyCF computation for $n = 2$ and $n = 6$.

All the above thresholding scheme are performed on a low-dimensional dataset.

## 4 THE CLOE METHOD

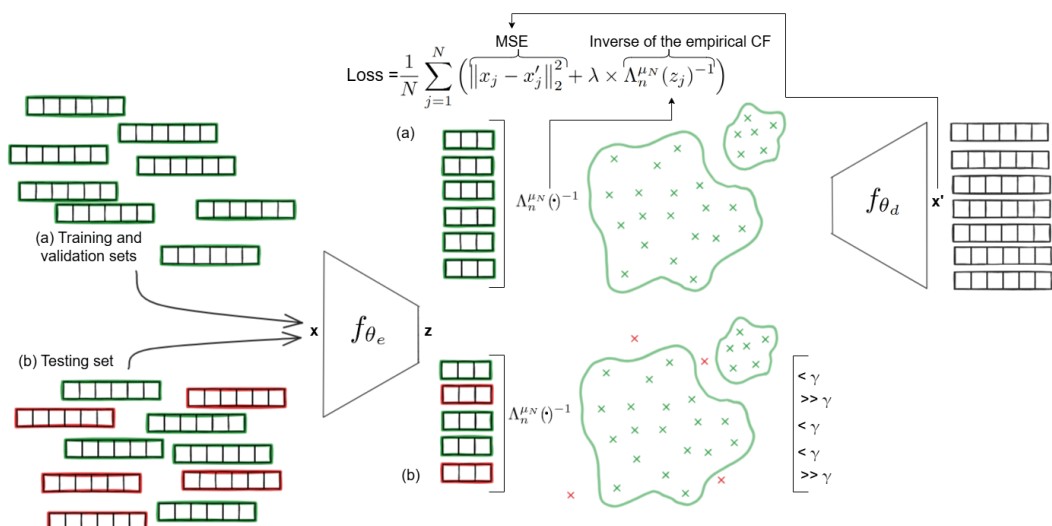

Figure 1: (a) Graphical representation of the joint training step of CLOE. The autoencoder, $f_{\theta_e}$ and $f_{\theta_d}$, is trained with normal data (green samples in the figure) to minimize the reconstruction error regularized by the inverse of the empirical CF computed on the latent space. The support of the latent space distribution is estimated with the empirical CF for all data points. (b) Graphical representation of the outlier detection step. Data points outside the support (in red) have CF values that increase exponentially with the hyperparameter $n$, much higher than the threshold $\gamma$, they are labeled as outliers.

CLOE ("Christoffel LOss for autoEncoder") is proposed as a method to utilize the inverse of the CF to detect outliers in high-dimensional datasets. CLOE jointly learns a new representation of the dataset in a low-dimensional space with an AE, regularized using the empirical CF in latent space. The proposed method has four different steps. The first three steps are dedicated to the training steps, their pseudo-algorithms are detailed in Appendix 7, Algorithm 1. The last step corresponds to the inference or anomaly detection step, its pseudo-algorithm is detailed in Appendix 7, Algorithm 2.

Let $\mathbb{X}_{train}$ be the training set, $\mathbb{X}_{valid}$ be the validation set, and $\mathbb{X}_{test}$ be the testing set. The training and validation sets contain only normal samples (in green on Figure 1). Let $f_{\theta_e} : \mathbb{X} \subset \mathbb{R}^D \to \mathbb{X}_e \subset \mathbb{R}^d$ and $f_{\theta_d} : \mathbb{X}_e \to \mathbb{R}^D$ be the encoder and the decoder neural networks, where $\theta_e$ and $\theta_d$ are learnable parameters. Let $\mathbb{X}_{e_{train}} = f_{\theta_e}(\mathbb{X}_{train})$ be the encoded training set, $\mathbb{X}_{e_{valid}} = f_{\theta_e}(\mathbb{X}_{valid})$ be the encoded validation set, and $\mathbb{X}_{e_{test}} = f_{\theta_e}(\mathbb{X}_{test})$ be the encoded testing set.

The AE training is divided into two parts, corresponding to the first two parts of the three training steps. Then the final training step corresponds to the support computation with the whole encoded training dataset and the definition of a threshold.

**Training part 1: pretraining.** The AE is trained only for reconstruction to initialize the network weights. The loss function is the Mean Square Error (MSE):

$$MSE : \frac{1}{N} \sum_{j=1}^{N} \left\| x_j - f_{\theta_d}\left(f_{\theta_e}\left(x_j\right)\right) \right\|_2^2 = \frac{1}{N} \sum_{j=1}^{N} \left\| x_j - x'_j \right\|_2^2 . \tag{10}$$

**Training part 2: joint training.** The joint training step of the model is illustrated in Figure 1 (a). The training of the AE is completed with a regularized loss that combines the reconstruction loss

with an empirical CF-based loss:

$$Loss : \frac{1}{N} \sum_{j=1}^{N} \left( \|x_j - f_{\theta_d}(f_{\theta_e}(x_j))\|_2^2 + \lambda \times \Lambda_n^{\mu_N}(f_{\theta_e}(x_j))^{-1} \right) \tag{11}$$

$$= \frac{1}{N} \sum_{j=1}^{N} \left( \|x_j - x_j'\|_2^2 + \lambda \times \Lambda_n^{\mu_N}(z_j)^{-1} \right), \tag{12}$$

where $\lambda$ is a dynamic regularization term that controls the strength of the Christoffel loss term. $\lambda$ is computed at each epoch as the quotient of the gradient norm of the MSE loss and the gradient norm of the CF loss when this latter gradient is non-zero. The support of the training dataset is computed using 80% of each batch training dataset, denoted $\mathbb{X}_{e_{train}}$. To obtain a good estimation of the support, the number of data points to compute the support must be at least $s_d(n)$ (Lasserre et al., 2022). After the support estimation, the CF is computed for all the training data and the mean of these values is utilized in the loss. As only normal data are used for the training, the value of the CF on those data should be close to 0. Adding this Christoffel-guided loss to the main loss of the AE and minimizing it helps the AE to learn representations more suitable for support estimation.

To have lower computational complexity and more stability, the Cholesky inversion method is used to invert the moment matrix. This matrix is positive definite, with the condition on $n$ and the size defined for the batch, singularity of this matrix does not need to be checked before inversion. To avoid instability during this inversion due to large values in the $\mathbb{X}_e$, data are normalized between $[-1, 1]^d$ at the end of the encoder, $\mathbb{X}_e \subseteq [-1, 1]^d$.

**Process to choose the hyperparameter $n$.** A validation step is performed at the end of each epoch. The support of the distribution is computed with all the training data, then the CF value of each sample of the validation set is computed. The mean of all the CF values of the validation set is used to compute the validation loss adding to the reconstruction loss. This validation loss is monitored after the first five epochs, and the value of $n$ is validated if the validation loss decreased during training for the following epochs. If the loss does not decrease, the value of $n$ should be changed to $n - 1$.

**Training part 3: Final support computing and threshold estimation**. The last step of the training step is to encode the full training set. Then, the support of the CF is computed. A new $n_{support} \geq n$ is chosen according to the condition that $s_{n_{support}}(d) < |\mathbb{X}_{e_{train}}|$. Then the threshold is set as:

$$\gamma_n = \max\{\Lambda_{n_{support}}^{\mu_N}(z)^{-1}, z \in \mathbb{X}_{e_{train}}\} \tag{13}$$

**Inference / anomaly detection**. Figure 1 (b) proposes a graphical representation of this step. For a new test sample $x_{test}$, compute its latent representation $z_{test} = f_{\theta_e}(x_{test})$ and Christoffel value $\Lambda_{n_{support}}^{\mu_N}(z_{test})^{-1}$. If $\Lambda_{n_{support}}^{\mu_N}(z_{test})^{-1} \leq \gamma_n$, then $x_{test}$ is an inlier; otherwise, $x_{test}$ is an outlier.

## 5 EXPERIMENTS

### 5.1 DATASETS

To evaluate the CLOE method, we use several datasets from ADBench (Han et al., 2022). This benchmark provides a diverse collection of datasets for anomaly detection with distinctive features. As our focus is on high dimensional data and not only images, we selected 15 datasets, each with 9 or more dimensions. The number of data points per dataset varies between 80 and 299285. Detailed characteristics of the selected datasets are presented in Appendix B, Table 6.

For each dataset, outliers are utilized exclusively during the testing step. The inlier dataset is split into a training (70%), validation (20%) and testing (10%) set. To compare our results to different baseline methods, we fix a random seed to produce identical splits across experiments.

### 5.2 BASELINE METHODS

Our method is compared to DAGMM (Zong et al., 2018), Og coupled version (Pinon & Lartizien, 2025), DRL (Ye et al., 2025), RCA (Liu et al., 2021), MCM (Yin et al., 2024), OC-SVM (Schölkopf

et al., 1999), iForest (Liu et al., 2008), ECOD (Li et al., 2022), DeepSVDD (Ruff et al., 2018), kNN (Ramaswamy et al., 2000) and KDE (Parzen, 1962). For DAGMM, we use the implementation proposed by Han et al. (2022). For Og, we use the implementation proposed in Pinon & Lartizien (2025) with PyTorch for their experiment number one. The implementation of the AE is modified with linear layers instead of two-dimensional convolutional layers. Models are trained for 400 epochs. For DRL, we use the implementation proposed by Ye et al. (2025). For RCA, we use the implementation proposed by Liu et al. (2021). For MCM, we use the implementation proposed by Yin et al. (2024). Then, for the last six models, we use the PyOD implementations (Zhao et al., 2019). The hyperparameters of all baselines are set according to the corresponding original papers, Appendix E Table 9 summarizes the hyperparameters for all the baseline methods.

DRL (Ye et al., 2025) is a state-of-the-art method for AD in high-dimensional tabular data. Unlike AE-based approaches, it constructs a new representation for the data using a feature extractor and uses a reconstruction loss to determine if the sample is an outlier.

MCM (Yin et al., 2024) learns intrinsic correlation in normal data, training a generator to mask the input sample. Then using an AE, it learns to reconstruct the input sample from the masked input sample. The model is trained with a reconstruction loss for the AE and with a diversity loss that encourages the generator to create masks that focus on diverse correlations existing in normal data.

RCA (Liu et al., 2021) is an AE-based approach to learn a reconstruction error. Many AEs trained with mini-batch are considered. For each mini-batch, the samples with the lowest reconstruction error in an AE are selected and used in the back-propagation step of the other AEs. Then the means of reconstructed errors of all the AEs are considered to determine if a sample is an outlier.

DAGMM (Zong et al., 2018) and Og (Pinon & Lartizien, 2025) are the methods most similar to CLOE. However, DAGMM uses a neural network to predict the sample mixture membership. The model is an adaptation of the mixture model. It differs from CLOE, which directly applies AD methods instead of adapting them.

Og (Pinon & Lartizien, 2025) does not consider a minimal value for the batch size to ensure a correct estimation of the support. Moreover, its non-differentiable loss can lead to gradient approximation issues during backpropagation. The main difference between Og and CLOE is that Og relies on OC-SVM to detect outliers from the support, whereas CLOE uses the empirical CF.

DeepSVDD (Ruff et al., 2018) is also a method with a deep neural network, similar to CLOE. However, the AE and the AD model are trained separately. The representations of the data may not be well-suited for SVDD.

OC-SCM (Schölkopf et al., 1999), iForest (Liu et al., 2008), ECOD (Li et al., 2022), kNN (Ramaswamy et al., 2000) and KDE (Parzen, 1962) are classical AD methods that do not rely on deep neural networks. They are computationally efficient but may struggle to achieve high performance on high-dimensional datasets.

### 5.3 EVALUATION METRICS

We evaluate our results using Area Under the Receiver Operating Characteristic curve (AU-ROC) and Average Precision Area Under Curve (AP AUC), the same metrics used in the ADBench paper (Han et al., 2022) to compare the different methods. Both metrics are computed using the implementation provided by the scikit-learn Python package (Pedregosa et al., 2011). The AU-ROC metric reflects the trade-off between true positive and false positive rates. AP AUC combines precision and recall metrics. It is particularly informative for imbalanced data, which is the case with all the datasets, as there are few outliers.

### 5.4 IMPLEMENTATION

CLOE is implemented with PyTorch[2]. As in Xie et al. (2016), the AE has 3 hidden layers of dimensions 500, 500, and 2000, using ReLU activation functions. The latent space dimension is set to $d = 8$, chosen according to the complexity of computing the moment matrix of the training set with $2 \leq n \leq 7$. A dropout rate of 20% is applied for the pretraining step and no dropout is used

---

[2]The code source is available in: the joint zip file

for the joint training step according to the configuration proposed in Xie et al. (2016). At the end of the encoder, a batchnorm layer followed by a Hyperbolic Tangent (Tanh) activation layer is added to ensure the encoded data lie within $[-1; 1]$. This condition is required to compute the moment matrix and invert it using the Cholesky algorithm.

The pretraining phase is conducted for 10 epochs with an early stopping rule based on the value of the validation loss. The joint training is conducted for 150 epochs with an early-stopping policy of 10 epochs. The Adam optimizer is used with a learning rate of $1e - 4$ for all datasets. All experiments were conducted on a device with 8 CPUs and 32 GB RAM. The training and inference times and the CPU memory required for training are detailed in Appendix C, Table 7.

The first two training steps are conducted in batches, with batch size set to $s_d(n)$, where $n \in \mathbb{N}$ is chosen so that the batch size is smaller than the number of data points used to compute the support in the training step: $s_n(d) < |\mathbb{X}_{e_{train}}| \times 0.8$. This ensures that at least one batch is large enough to compute the support during the joint training step. In Appendix D, Table 8 details the hyperparameter $n$ for each dataset.

## 5.5 RESULTS

Table 1: AU-ROC for the different methods on the selected datasets

| Dataset | CLOE | DAGMM | Og | DRL | RCA | MCM | OC-SVM | iForest | ECOD | Deep SVDD | kNN | KDE |
|---|---|---|---|---|---|---|---|---|---|---|---|---|
| ALOI | **0.561** | 0.529 | N/A | 0.523 | 0.546 | 0.534 | 0.517 | 0.539 | 0.531 | 0.546 | 0.556 | 0.518 |
| backdoor | **0.944** | 0.619 | N/A | 0.927 | 0.855 | 0.891 | 0.865 | 0.750 | 0.846 | 0.553 | 0.938 | 0.915 |
| breastw | 0.994 | N/A | 0.367 | 0.990 | 0.995 | 0.995 | 0.997 | 0.994 | 0.994 | 0.988 | 0.995 | **0.998** |
| campaign | 0.610 | 0.603 | N/A | **0.745** | 0.689 | 0.686 | 0.689 | 0.721 | 0.772 | 0.710 | 0.725 | 0.699 |
| cardio | **0.979** | 0.527 | N/A | 0.915 | 0.954 | 0.913 | 0.957 | 0.951 | 0.946 | 0.933 | 0.933 | 0.977 |
| census | 0.629 | 0.605 | N/A | 0.664 | 0.605 | 0.624 | 0.553 | 0.611 | 0.659 | **0.702** | 0.661 | 0.662 |
| fault | **0.928** | 0.496 | N/A | 0.797 | 0.679 | 0.716 | 0.591 | 0.662 | 0.485 | 0.542 | 0.822 | 0.884 |
| Hepatitis | **0.938** | 0.589 | 0.625 | 0.702 | 0.754 | 0.555 | 0.855 | 0.816 | 0.786 | 0.789 | 0.639 | 0.855 |
| InternetAds | **0.878** | N/A | N/A | 0.877 | 0.689 | 0.708 | 0.425 | 0.698 | 0.749 | | 0.823 | 0.815 |
| landsat | **0.854** | 0.580 | N/A | 0.819 | 0.593 | 0.603 | 0.471 | 0.614 | 0.388 | 0.462 | 0.784 | 0.757 |
| letter | 0.943 | 0.391 | N/A | 0.762 | 0.757 | 0.501 | 0.977 | 0.639 | 0.579 | 0.523 | 0.917 | **0.980** |
| mnist | 0.750 | 0.615 | N/A | **0.974** | 0.892 | 0.936 | 0.789 | 0.860 | 0.768 | 0.834 | 0.937 | 0.920 |
| musk | **1.0** | 0.485 | N/A | 0.999 | 0.999 | 0.997 | 0.859 | 0.960 | 0.993 | 0.998 | 1.0 | 1.0 |
| shuttle | **0.998** | 0.991 | N/A | 0.994 | 0.992 | 0.992 | 0.997 | 0.996 | 0.993 | 0.994 | 0.995 | 0.997 |
| speech | 0.859 | 0.489 | N/A | 0.667 | 0.472 | 0.486 | 0.469 | 0.479 | 0.473 | 0.508 | 0.501 | **0.881** |
| Mean | **0.858** | 0.578 | 0.496 | 0.823 | 0.765 | 0.746 | 0.753 | 0.734 | 0.727 | 0.723 | 0.815 | 0.857 |
| Rank | 1 | 11 | 12 | 3 | 5 | 6 | 7 | 8 | 9 | 10 | 4 | 2 |

Table 2: AP AUC for the different methods on the selected datasets

| Dataset | CLOE | DAGMM | Og | DRL | RCA | MCM | OC-SVM | iForest | ECOD | Deep SVDD | kNN | KDE |
|---|---|---|---|---|---|---|---|---|---|---|---|---|
| ALOI | 0.044 | 0.041 | N/A | 0.038 | 0.023 | 0.042 | 0.041 | 0.033 | 0.032 | 0.037 | **0.049** | 0.042 |
| backdoor | 0.745 | 0.033 | N/A | **0.792** | 0.102 | 0.281 | 0.107 | 0.048 | 0.093 | 0.038 | 0.517 | 0.411 |
| breastw | 0.985 | N/A | 0.204 | 0.978 | 0.991 | 0.991 | **0.994** | 0.989 | 0.987 | 0.973 | 0.991 | **0.996** |
| campaign | 0.178 | 0.177 | N/A | 0.285 | 0.270 | 0.266 | 0.310 | 0.302 | **0.356** | 0.290 | 0.304 | 0.296 |
| cardio | 0.817 | 0.116 | N/A | 0.739 | 0.723 | 0.587 | 0.665 | 0.679 | 0.626 | 0.705 | 0.667 | **0.861** |
| census | 0.084 | 0.086 | N/A | 0.094 | 0.070 | 0.077 | 0.065 | 0.074 | 0.084 | **0.126** | 0.084 | 0.084 |
| fault | **0.828** | 0.365 | N/A | 0.700 | 0.494 | 0.588 | 0.458 | 0.495 | 0.337 | 0.419 | 0.668 | 0.825 |
| Hepatitis | **0.670** | 0.214 | 0.361 | 0.335 | 0.434 | 0.216 | 0.395 | 0.400 | 0.356 | 0.439 | 0.251 | 0.747 |
| InternetAds | 0.526 | N/A | N/A | 0.668 | 0.501 | 0.596 | 0.578 | 0.155 | 0.552 | 0.495 | 0.692 | 0.747 |
| landsat | **0.739** | 0.267 | N/A | 0.637 | 0.246 | 0.272 | 0.199 | 0.273 | 0.172 | 0.195 | 0.473 | 0.499 |
| letter | 0.644 | 0.067 | N/A | 0.251 | 0.165 | 0.165 | 0.731 | 0.091 | 0.079 | 0.074 | 0.411 | 0.723 |
| mnist | 0.315 | 0.170 | N/A | **0.843** | 0.454 | 0.735 | 0.194 | 0.377 | 0.194 | 0.455 | 0.666 | 0.640 |
| musk | 0.999 | 0.048 | N/A | 0.990 | 0.982 | 0.978 | 0.104 | 0.472 | 0.855 | 0.941 | 0.999 | 0.999 |
| shuttle | 0.978 | 0.853 | N/A | 0.894 | 0.972 | 0.841 | 0.939 | 0.976 | 0.912 | 0.914 | 0.854 | 0.875 |
| speech | 0.068 | 0.016 | N/A | 0.044 | 0.019 | 0.024 | 0.019 | 0.079 | 0.020 | 0.017 | 0.020 | 0.118 |
| Mean | **0.575** | 0.189 | 0.283 | 0.553 | 0.429 | 0.438 | 0.387 | 0.363 | 0.377 | 0.408 | 0.510 | 0.569 |
| Rank | 1 | 12 | 11 | 3 | 6 | 5 | 8 | 10 | 9 | 7 | 4 | 2 |

Tables 1 and 2 show the results for the 15 selected datasets with the metric AU-ROC and AP AUC for CLOE and its baselines. Experiments were repeated 5 times with different random seeds, and the mean results are presented. The highest values are in bold and the second are underlined. Appendix J, Table 18, and Table 19 present the variances of the experiments. Entries marked as 'N/A' indicate that the model could not be trained on the corresponding dataset.

Regarding the deep learning methods, DAGMM requires a matrix that is not always invertible, preventing successful training on some datasets (marked as N/A in the tables). Training Og on datasets with more than 100 samples requires GPU acceleration. For datasets larger than one thousand samples, memory requirements exceed 30 GB, which is beyond our machine's capacity, resulting in additional 'N/A' entries. DRL was trained with GPU acceleration using a 4-GPU device (15.3 GB memory per GPU), although it can also run on the same 8-CPU device as CLOE. Across the test datasets, CLOE outperforms Og, DeepSVDD, RCA, MCM and DAGMM. Compared to DRL, CLOE achieves better performance on 12 datasets for AU-ROC and 10 datasets for AP AUC. No-

tably, no distance calculation is required to compute the anomaly score for CLOE, unlike DRL, and only one hyperparameter is required to train the method, compared to five for DRL.

Regarding classical AD methods, CLOE outperforms in 9 datasets according to AU-ROC and in 6 according to AP AUC. CLOE is on average better than the other classical methods on all datasets for both metrics. However, KDE and kNN obtain good performances on some datasets. As was shown in Ducharlet et al. (2024), the visual analysis of the level sets produced by the CF-based AD method and KDE shows that the level sets of the CF-based AD method are better fitted to data distribution.

## 5.6 ABLATION STUDIES

Table 3: AU-ROC for the ablation study

| Dataset | CLOE | Without pretraining | Without joint training | Untrained AE |
|---------|------|---------------------|------------------------|--------------|
| Mean | **0.858**($\pm$**0.021**) | 0.802($\pm$0.032) | 0.722($\pm$0.018) | 0.725($\pm$0.026) |

Table 4: AP AUC for the ablation study

| Dataset | CLOE | Without pretraining | Without joint training | Untrained AE |
|---------|------|---------------------|------------------------|--------------|
| Mean | **0.575**($\pm$**0.114**) | 0.504($\pm$0.108) | 0.369($\pm$0.070) | 0.394($\pm$0.083) |

To check the utility of each training step of the AE of our method, we performed an ablation study using all datasets.

First, we removed the pretraining step. The weights of the AE are randomly initialized and the joint training is performed until the validation loss stops improving. The joint training and the support computation steps remain unchanged from the full method.

Second, we removed the joint training step. The AE is first trained for 10 epochs using only the reconstruction loss (Equation 10). The CF support is then computed and the threshold defined in the original method is used.

Then, (Ryu et al., 2024) raised a warning concerning good performance of untrained neural network. To confirm the utility to train the AE in CLOE, an experiment with randomly initialized weights and data encoded from the latent space of the untrained AE is conducted. The CF is trained with these encoded data.

Results are presented in Tables 3 for the AU-ROC metric and 4 for AP AUC metric, detail results for all the dataset are presented in Appendix G, Table 14 and Table 15. The study shows that pretraining step and joint training are needed, as models without pretraining or without joint training step underperform compared to CLOE. For datasets of dimension 9, the performance without the joint training step is very close to CLOE performance. These results confirm that CLOE is designed for high-dimensional tabular datasets. CLOE is recommended for dimensions higher than 10. For lower dimensions, the recommendation is for CF-based AD method without AE (Ducharlet et al., 2024). The untrained AE can obtain good performance on some datasets like *shuttle* or *campaign*. A complete training strategy improves the mean AU-ROC and AP AUC scores by approximately 7% to 16% and 12% to 36%, respectively. This highlights the importance of implementing the complete training approach.

Finally, to confirm our thresholding scheme, an experiment has been conducted with different methods to determine the threshold, using the F1-score to compare the results (cf. Table 5). The results of the "Optimized" column are obtained with a threshold iteratively optimized on the F1-score. Those of the "Adjusted" column are obtained with a threshold adjusted on the outlier contamination ratio of the test dataset. The "CLOE" column reports the results with a threshold indicated by our method. Finally, the other columns report the results for the thresholds set by quartiles of the training or validation sets, specifically 50th (median), 75th, 90th, and 100th percentiles.

Table 5: F1-score for different threshold

| Dataset | Optimized | Adjusted | CLOE | 90th p train | 75th p train | 50th p train | 100th p valid | 90th p valid | 75th p valid | 50th p valid |
|---|---|---|---|---|---|---|---|---|---|---|
| *ALOI* | 0.075 | 0.066 | **0.075** | 0.072 | 0.065 | 0.064 | 0.010 | 0.073 | 0.067 | 0.064 |
| *backdoor* | 0.413 | 0.334 | 0.262 | 0.233 | 0.138 | 0.083 | **0.411** | 0.241 | 0.147 | 0.083 |
| *breastw* | 0.949 | 0.941 | 0.937 | 0.880 | 0.793 | 0.670 | 0.222 | **0.950** | 0.871 | 0.710 |
| *campaign* | 0.248 | 0.194 | 0.231 | 0.245 | 0.236 | 0.223 | 0.008 | 0.186 | 0.233 | **0.246** |
| *cardio* | 0.681 | 0.681 | 0.432 | 0.377 | 0.315 | 0.25 | 0.390 | **0.690** | 0.636 | 0.553 |
| *census* | 0.170 | 0.098 | 0.119 | 0.143 | 0.157 | **0.167** | 0.007 | 0.125 | 0.155 | **0.167** |
| *fault* | 0.811 | 0.765 | **0.793** | 0.752 | 0.699 | 0.625 | 0.017 | 0.454 | 0.610 | 0.779 |
| *Hepatitis* | 0.720 | 0.692 | 0.565 | 0.520 | 0.448 | 0.388 | 0.133 | 0.571 | 0.440 | **0.667** |
| *InternetAds* | 0.632 | 0.522 | **0.621** | 0.562 | 0.500 | 0.418 | 0.016 | 0.393 | 0.496 | 0.549 |
| *landsat* | 0.741 | 0.708 | **0.732** | 0.652 | 0.562 | 0.461 | 0.178 | 0.585 | 0.688 | 0.728 |
| *letter* | 0.405 | 0.380 | 0.323 | 0.278 | 0.227 | 0.174 | 0.019 | 0.352 | 0.362 | **0.382** |
| *mnist* | 0.253 | 0.208 | 0.243 | **0.246** | 0.221 | 0.194 | 0.0 | 0.158 | 0.224 | 0.234 |
| *musk* | 1.0 | 1.0 | 0.951 | 0.381 | 0.206 | 0.115 | **0.979** | 0.421 | 0.207 | 0.114 |
| *shuttle* | 0.980 | 0.976 | **0.938** | 0.501 | 0.338 | 0.222 | 0.912 | 0.576 | 0.368 | 0.235 |
| *speech* | 0.107 | 0.032 | **0.107** | 0.087 | 0.068 | 0.049 | 0.0 | 0.044 | 0.068 | 0.096 |
| Mean | 0.546 | 0.506 | **0.488** | 0.395 | 0.332 | 0.380 | 0.220 | 0.349 | 0.385 | 0.374 |

On average across all datasets, the CLOE threshold achieves the best performance after the Adjusted threshold. This study shows the robustness of CLOE to determine automatically the threshold.

## 6 CONCLUSION AND FUTURE WORKS

In this work, we propose CLOE, an empirical CF guided AE method, to detect outliers in high-dimensional data. Importantly, CLOE requires tuning of only one single hyperparameter. One limitation of CLOE is that it requires a reduced dimension of the latent space, set to 8 in this work, due to the increasing size of the moment matrix to invert. The experiments show that CLOE obtains outstanding results for most of the dataset. For the highest dimensional dataset, CLOE is the most efficient AD method. In addition, CLOE comes with an automatic threshold scheme that provides a robust way to detect outliers. Interestingly CLOE is designed to be trained without a GPU.

## 7 REPRODUCIBILITY STATEMENT

The code of the proposed methods is joined in a zip file, it will be available on GitLab after the anonymous review step. All the tests conducted in this paper can be reproduced, with CLOE and with the baseline methods. The READ ME file explains how to use the code.

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

# A   ALGORITHM

---

**Algorithm 1** Training of CLOE

---

    **Input:** $n, n_{support}, \mathbb{X}_{train}, \mathbb{X}_{valid}, epoch_{pre}, epoch_{join}$
    **Output:** Trained autoencoder $f_{\theta_e}$, trained CF $\Lambda_n^{\mu_N}(.)^{-1}$, threshold $\gamma_{n_{support}}$
$d \leftarrow$ dimension of the latent space
$bs \geq \binom{d+n}{n}$                                                           ▷ Batch size
$N \leftarrow |\mathbb{X}_{train}|$                                                    ▷ Size of the training set
**Pretraining**:
**for** each $epoch_{pre}$ **do**
    **for** every batch of training samples $(x_{i_t})_{1 \leq i \leq bs}$ **do**
        Compute the reconstruction of the sample: $x'_{i_t} = f_{\theta_d}(f_{\theta_e}(x_{i_t})), \forall i \in [1, bs]$
        Compute the MSE loss (Equation 10)
        Apply gradient step to $f_{\theta_e}$ and $f_{\theta_d}$
    **end for**
**end for**
**Joint training:**
**for** each $epoch_{join}$ **do**
    **for** every batch of training samples $(x_{i_t})_{1 \leq i \leq bs}$ **do**
        Compute the latent representation, $z_{i_t} = f_{\theta_e}(x_{i_t}), \forall i \in [1, bs]$
        Compute the reconstruction of the latent space, $x'_{i_t} = f_{\theta_d}(z_{i_t}), \forall i \in [1, bs]$
        Split the $(z_{i_t})_{1 \leq i \leq bs}$ in 80% $(z_{i_t}^{80})_{1 \leq i \leq 0.8bs}$ and 20% $(z_{i_t}^{20})_{1 \leq i \leq 0.2bs}$ sets
        Compute the support of the CF $\mu_{0.8bs}$ with $(z_{i_t}^{80})_{1 \leq i \leq 0.8bs}$ set
        Compute the CF value $(\Lambda_n^{\mu_{0.8bs}}(z_{i_t})^{-1})_{1 \leq i \leq bs}$ for $(z_{i_t})_{1 \leq i \leq bs}$
        Compute the loss (Equation 11) and apply gradient step to $f_{\theta_e}$ and $f_{\theta_d}$
    **end for**
    Compute the latent representation of every training sample $(z_{i_t})_{1 \leq i \leq N}, z_{i_t} = f_{\theta_e}(x_{i_t}), \forall i \in [1, N]$
    Compute the support of the CF $\mu_N$ with $(z_{i_t})_{1 \leq i \leq N}$
    **for** each validation sample $x_v$ **do**
        Compute the latent representation of $x_v$, $z_v = f_{\theta_e}(x_v)$
        Compute the reconstruction of latent representation of sample $x_v$, $x'_v = f_{\theta_d}(z_v)$
        Compute the CF value of $z_v$, $\Lambda_n^{\mu_N}(z_v)^{-1}$
        Compute the loss (Equation 11)
    **end for**
    Display the mean of all the validation loss
**end for**
**if** The validation does not decreased through the epochs **then**
    Stop training
    $n \longleftarrow n - 1$
    Start training again from pretraining step
**end if**
**Final support computing and threshold estimation:**
Compute the latent representation of each training sample $(z_{i_t})_{1 \leq i \leq N}, z_{i_t} = f_{\theta_e}(x_{i_t}), \forall i \in [1, N]$
Compute the support of the CF $\mu_N$ with $(z_{i_t})_{1 \leq i \leq N}$
Compute the CF value of each training sample $(z_t)_{1 \leq i \leq N}, (\Lambda_{n_{support}}^{\mu_N}(z_{i_t})^{-1})_{1 \leq i \leq N}$
$\gamma_{n_{support}} \leftarrow \max_{1 \leq i \leq N}(\Lambda_{n_{support}}^{\mu_N}(z_{i_t})^{-1})$
**return** $f_{\theta_e}, \Lambda_n^{\mu_N}(.)^{-1}, \gamma_{n_{support}}$

---

**Algorithm 2** Inference and outlier detection with CLOE

**Input:** Trained autoencoder $f_{\theta_e}$, trained CF $\Lambda_n^{\mu_N}(.)^{-1}$, threshold $\gamma_{n_{support}}$, test sample $x_{test}$
**Output:** 0 (inlier) or 1 (outlier)
Compute the CF value of each testing sample $(z_{test})_{1 \leq i \leq N}, (\Lambda_{n_{support}}^{\mu_N}(z_{i_t})^{-1})_{1 \leq i \leq N}$
**if** $\Lambda_{n_{support}}^{\mu_N}(z_{test})^{-1} \leq \gamma_{n_{support}}$ **then**
    **return** 0                                               $\triangleright$ $x_{test}$ is an inlier
**else**
    **return** 1                                               $\triangleright$ $x_{test}$ is an outlier
**end if**

## B   DATASETS DETAILS FOR THE EXPERIMENTS

Table 6: Details of the chosen datasets

| Dataset | Number data | Number of Features | % outlier | Category |
|---|---|---|---|---|
| *ALOI* | 49534 | 27 | 3.04 | Image |
| *backdoor* | 95329 | 196 | 2.44 | Network |
| *breastw* | 683 | 9 | 34.99 | Healthcare |
| *campaign* | 41188 | 62 | 11.27 | Finance |
| *cardio* | 1831 | 21 | 9.61 | Healthcare |
| *census* | 299285 | 500 | 6.2 | Sociology |
| *fault* | 1941 | 27 | 34.67 | Physics |
| *Hepatitis* | 80 | 19 | 16.25 | Healthcare |
| *InternetAds* | 1966 | 1555 | 18.72 | Image |
| *landsat* | 6435 | 36 | 20.71 | Astronautics |
| *letter* | 1600 | 32 | 6.25 | Image |
| *mnist* | 7603 | 100 | 9.21 | Image |
| *musk* | 3062 | 166 | 3.17 | Chemistry |
| *shuttle* | 49097 | 9 | 7.15 | Astronautics |
| *speech* | 3686 | 400 | 1.65 | Linguistics |

## C   MEMORY AND TIME FOR TRAINING AND INFERENCE ON A CPU

In this section, we provide for each dataset the training time of CLOE, the CPU memory usage, the time needed to infer the whole dataset (Inference time), and the time needed to infer a single sample.

Table 7: Memory and time for training and inference on a CPU

| Dataset | Training time (s) | CPU Memory for training (Mb) | Inference time (s) | Inference time for one sample (s) |
|---|---|---|---|---|
| *ALOI* | 3213 | 1862 | 76 | 2e-5 |
| *backdoor* | 3075 | 2253 | 292 | 3e-3 |
| *breastw* | 158 | 828 | 4.7 | 6.8e-3 |
| *campaign* | 3216 | 1866 | 127 | 3e-3 |
| *cardio* | 731 | 1049 | 3.79 | 4.1e-3 |
| *census* | 2324 | 5229 | 921 | 3e-3 |
| *fault* | 514 | 1026 | 4.59 | 4.-3 |
| *Hepatitis* | 33 | 809 | 0.013 | 4e-3 |
| *InternetAds* | 549 | 1371 | 8.5 | 4e-3 |
| *landsat* | 3134 | 1207 | 124 | 1.9e-2 |
| *letter* | 660 | 1056 | 7.8 | 4e-3 |
| *mnist* | 3515 | 1232 | 167 | 2e-2 |
| *musk* | 1995 | 1171 | 14 | 4e-3 |
| *shuttle* | 2434 | 1911 | 162 | 3e-3 |
| *speech* | 720 | 1354 | 5.6 | 2e-3 |

# D TRAINING HYPERPARAMETERS FOR CLOE

In Table 8, the hyperparameter $n$ is given for each dataset. The parameter $n_{support}$ is computed according to the heuristic proposed by Vu et al. (2022) at Section 4.1.

Table 8: Training hyperparameters of CLOE for the different datasets

| Dataset | $n$ (joint training step) | Computed $n_{support}$ |
|---|---|---|
| ALOI | 5 | 6 |
| backdoor | 5 | 6 |
| breastw | 4 | 5 |
| campaign | 5 | 6 |
| cardio | 4 | 5 |
| census | 5 | 5 |
| fault | 5 | 5 |
| Hepatitis | 2 | 2 |
| InternetAds | 4 | 5 |
| landsat | 4 | 6 |
| letter | 4 | 5 |
| mnist | 4 | 5 |
| musk | 4 | 5 |
| shuttle | 5 | 6 |
| speech | 4 | 6 |

# E DETAILS ABOUT HYPERPARAMETERS

Table 9: Hyperparameter values used for the baseline methods

| Method | Hyperparameters | Values |
|---|---|---|
| **Og** | OC SVM coefficient | 0.1 |
| | OC SVM $\nu$ coefficient | 0.03 |
| | $\gamma$ radial basis function coefficient | scale |
| | Learning rate | 1e-3 |
| | Latent dimension | 32 |
| | Epochs number | 400 |
| **DAGMM** | GMM number | 5 |
| | Lambda cov | 0.005 |
| | Learning rate | 1e-4 |
| | Latent dimension | 1 |
| | Epochs number | 400 |
| **DRL** | Diversity | True |
| | Plearn | False |
| | Input info ration | 0.1 |
| | Cl ration | 0.06 |
| | Basis vector num | 5 |
| | Learning rate | 0.05 |
| | Latent dimension | 128 |
| | Epochs number | 200 |
| **MCM** | Mask number | 15 |
| | $\lambda$ | 5 |
| | $\tau$ | 0.1 |
| | Learning rate | 0.05 |
| | Latent dimension | 128 |
| | Epochs number | 200 |
| **RCA** | AEs number | 2 |
| | Learning rate | 3e-4 |
| | Latent dimension | 256 |
| | Epochs number | 200 |
| **OC-SVM** | kernel | radial basis function |
| | $\nu$ coefficient | 0.5 |
| | $\gamma$ | scale |
| **iForest** | Estimators number | 100 |
| | Maximum of features | 1 |
| **Deep-SVDD** | Deep SVDD center | forward_nn_pass |
| | Use AE | False |
| | Optimizer | Adam |
| | Hidden layer dimensions | [64, 32] |
| | Epochs number | 100 |
| **kNN** | Neighbor number | 5 |
| | Method | largest |
| | Radius | 1.0 |
| | Leaf size | 30 |
| | Metric | Minkowski |
| | Parameter for Minkowski | 2 |
| | Algorithm | auto |
| **KDE** | Bandwidth | 1.0 |
| | Algorithm | Auto |
| | Leaf size | 30 |
| | Metric | Minkowski |

## F  TEST PARAMETERS

In this section, we analyze the parameters of CLOE. Experiments were carried out on two datasets, *Hepatitis* and *letter*. Several values were tested for the number of epochs used to pretrain the autoencoder (Tables 10 and 11) and for the learning rate (Tables 12 and 13). These tests validate the parameter choices adopted for CLOE. The highest values are in bold.

Table 10: AU-ROC for different number of epochs to pretrain the autoencoder

| Dataset | 10 | 50 | 100 | 200 | 500 |
|---|---|---|---|---|---|
| *Hepatitis* | **0.938** | 0.927 | 0.902 | 0.925 | 0.931 |
| *letter* | **0.943** | 0.936 | 0.929 | 0.926 | 0.912 |

Table 11: AP AUC for different number of epochs to pretrain the autoencoder

| Dataset | 10 | 50 | 100 | 200 | 500 |
|---|---|---|---|---|---|
| *Hepatitis* | **0.670** | 0.608 | 0.492 | 0.611 | 0.648 |
| *letter* | **0.644** | 0.520 | 0.427 | 0.427 | 0.359 |

Table 12: AU-ROC for different learning rate values to train CLOE

| Dataset | 1e-2 | 1e-3 | 1e-4 | 1e-5 |
|---|---|---|---|---|
| *Hepatitis* | 0.912 | 0.920 | **0.938** | 0.891 |
| *letter* | 0.694 | 0.797 | **0.943** | 0.898 |

Table 13: AP AUC for different learning rate values to train CLOE

| Dataset | 1e-2 | 1e-3 | 1e-4 | 1e-5 |
|---|---|---|---|---|
| *Hepatitis* | 0.584 | 0.622 | **0.670** | 0.508 |
| *letter* | 0.0.187 | 0.291 | **0.644** | 0.603 |

## G  DETAILS OF THE RESULTS OF THE ABLATION STUDY ON ALL DATASETS

In this section, Table 14 and Table 15 present the detailed ablation study results for all the datasets for the AU-ROC and AP AUC metrics. The highest values are in bold.

Table 14: AU-ROC for the ablation study

| Dataset | CLOE | Without pretraining | Without joint training | Untrained AE |
|---------|------|---------------------|------------------------|--------------|
| *ALOI* | 0.561 | **0.577** | 0.554 | 0.549 |
| *backdoor* | **0.944** | 0.922 | 0.851 | 0.826 |
| *breastw* | **0.994** | **0.994** | 0.929 | 0.986 |
| *campaign* | **0.610** | 0.439 | 0.561 | 0.604 |
| *cardio* | **0.979** | 0.904 | 0.785 | 0.874 |
| *census* | **0.629** | 0.544 | 0.624 | 0.611 |
| *fault* | **0.928** | 0.875 | 0.816 | 0.616 |
| *Hepatitis* | **0.938** | 0.901 | 0.618 | 0.802 |
| *InternetAds* | **0.878** | 0.862 | 0.725 | 0.625 |
| *landsat* | **0.854** | 0.812 | 0.780 | 0.608 |
| *letter* | **0.943** | 0.936 | 0.738 | 0.557 |
| *mnist* | **0.750** | 0.623 | 0.637 | 0.731 |
| *musk* | **1.0** | **1.0** | 0.856 | 0.929 |
| *shuttle* | **0.998** | 0.994 | 0.883 | 0.995 |
| *speech* | **0.859** | 0.650 | 0.463 | 0.508 |
| Mean | **0.858** | 0.802 | 0.722 | 0.725 |

Table 15: AP AUC for the ablation study

| Dataset | CLOE | Without pretraining | Without joint training | Untrained AE |
|---------|------|---------------------|------------------------|--------------|
| *ALOI* | **0.044** | 0.040 | 0.039 | 0.038 |
| *backdoor* | **0.745** | 0.544 | 0.202 | 0.454 |
| *breastw* | 0.985 | **0.988** | 0.854 | 0.975 |
| *campaign* | **0.178** | 0.103 | 0.141 | 0.169 |
| *cardio* | **0.817** | 0.565 | 0.398 | 0.584 |
| *census* | 0.084 | 0.065 | **0.088** | 0.077 |
| *fault* | **0.828** | 0.792 | 0.728 | 0.478 |
| *Hepatitis* | **0.670** | 0.550 | 0.363 | 0.477 |
| *InternetAds* | **0.526** | 0.522 | 0.386 | 0.330 |
| *landsat* | **0.739** | 0.618 | 0.572 | 0.315 |
| *letter* | **0.644** | 0.512 | 0.234 | 0.114 |
| *mnist* | 0.315 | 0.261 | 0.192 | **0.329** |
| *musk* | **0.999** | 0.998 | 0.522 | 0.616 |
| *shuttle* | **0.978** | 0.956 | 0.794 | 0.938 |
| *speech* | **0.068** | 0.043 | 0.017 | 0.024 |
| Mean | **0.575** | 0.504 | 0.369 | 0.394 |

## H  Monitoring of training losses

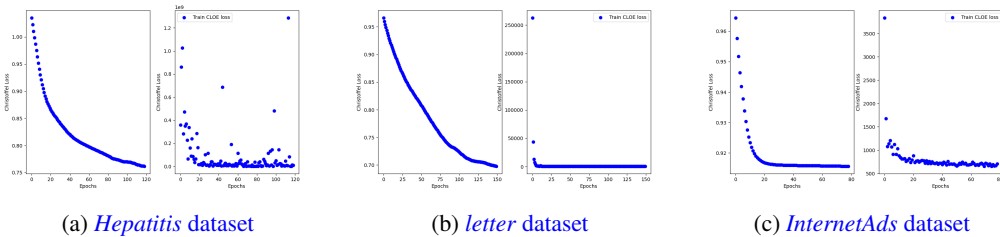

(a) *Hepatitis* dataset      (b) *letter* dataset      (c) *InternetAds* dataset

Figure 2: MSE loss (left) and CF loss (right) for different datasets

To confirm the impact of both losses during the joint training step, losses were monitored during this step. This section shows the evolution of the MSE loss and the CF loss across epochs for three datasets in Figure 2. Both losses decreased, indicating that each contributes effectively during the joint training.

## I  Experiment on real data

Table 16: AU-ROC for the different methods on real datasets

| Dataset | CLOE | DRL | RCA | MCM | OC-SVM | iForest | ECOD | Deep SVDD | kNN | KDE |
|---|---|---|---|---|---|---|---|---|---|---|
| Dataset 1 | 1.0 | 0.997 | 0.627 | 0.992 | 0.998 | 0.998 | 0.983 | 0.999 | 0.998 | 0.998 |
| Dataset 2 | 1.0 | 1.0 | 0.956 | 0.991 | 1.0 | 1.0 | 0.998 | 0.999 | 1.0 | 1.0 |
| Dataset 3 | 1.0 | 1.0 | 0.975 | 0.992 | 1.0 | 0.998 | 0.996 | 1.0 | 1.0 | 1.0 |
| Dataset 4 | 1.0 | 0.997 | 0.665 | 0.982 | 0.999 | 0.998 | 0.984 | 0.999 | 0.999 | 0.999 |
| Dataset 5 | 1.0 | 1.0 | 0.656 | 0.995 | 1.0 | 1.0 | 0.998 | 1.0 | 1.0 | 1.0 |
| Dataset 6 | 1.0 | 1.0 | 0.592 | 0.957 | 1.0 | 0.998 | 0.994 | 0.999 | 1.0 | 1.0 |
| Mean | **1** | 0.999 | 0.745 | 0.985 | 0.9995 | 0.9987 | 0.9925 | 0.9993 | 0.9995 | 0.9995 |

Table 17: AP AUC for the different methods on real datasets

| Dataset | CLOE | DRL | RCA | MCM | OC-SVM | iForest | ECOD | Deep SVDD | kNN | KDE |
|---|---|---|---|---|---|---|---|---|---|---|
| Dataset 1 | 0.996 | 0.914 | 0.257 | 0.645 | 0.991 | 0.998 | 0.676 | 0.923 | 0.991 | 0.992 |
| Dataset 2 | 1.0 | 0.999 | 0.749 | 0.753 | 0.999 | 0.995 | 0.998 | 0.999 | 1.0 | 0.999 |
| Dataset 3 | 0.997 | 0.990 | 0.621 | 0.740 | 0.994 | 0.946 | 0.872 | 0.994 | 0.994 | 0.995 |
| Dataset 4 | 0.998 | 0.988 | 0.570 | 0.698 | 0.987 | 0.938 | 0.787 | 0.982 | 0.983 | 0.984 |
| Dataset 5 | 0.999 | 0.998 | 0.278 | 0.833 | 0.999 | 0.973 | 0.907 | 0.999 | 0.999 | 0.999 |
| Dataset 6 | 0.999 | 0.993 | 0.229 | 0.524 | 0.998 | 0.943 | 0.793 | 0.997 | 0.998 | 0.998 |
| Mean | **0.998** | 0.980 | 0.451 | 0.699 | 0.995 | 0.966 | 0.839 | 0.982 | 0.994 | 0.996 |

An experiment was conducted on real data using six different tabular datasets. Each dataset has dimension 824 and contains between 60000 and 85000 samples. The percentage of outliers is very low, around 3% for each dataset. All datasets have been preprocessed before trainings to have zero mean and unit variance. For both CLOE and the baseline models, 5000 samples from each dataset have been used for training. The baseline implementations and hyperparameters are the same as in Section 5. For CLOE, hyperparameter $n$ is fixed to 4. Results are reported in Table 16 for AU-ROC and in Table 17 for AP AUC. All the methods obtain very good performances, but CLOE is the only one that reaches perfect AU-ROC on all datasets and obtains the best mean for the AP AUC.

## J  Variance of the experiments

5 runs were conducted for each method on each public dataset. This appendix provides the complete table with the detailed variances: Table 18 for AU-ROC in and Table 19 for the AP AUC.

Table 18: AU-ROC for the different methods on the selected datasets, with variances

| Dataset | CLOE | DAGMM | Og | DRL | RCA | MCM | OC-SVM | iForest | ECOD | DeepSVDD | kNN | KDE |
|---|---|---|---|---|---|---|---|---|---|---|---|---|
| ALOI | 0.561 ($\pm$3e-5) | 0.529 ($\pm$1e-4) | N/A | 0.523 ($\pm$8e-5) | 0.546 ($\pm$2e-4) | 0.534 ($\pm$2e-5) | 0.517 ($\pm$2e-8) | 0.539 ($\pm$6e-6) | 0.531 ($\pm$1e-9) | 0.546 ($\pm$1e-4) | 0.556 ($\pm$4e-6) | 0.518 ($\pm$1e-8) |
| backdoor | 0.944 ($\pm$1e-2) | 0.619 ($\pm$3e-3) | N/A | 0.927 ($\pm$2e-5) | 0.855 ($\pm$3e-5) | 0.891 ($\pm$1e-3) | 0.865 ($\pm$3e-6) | 0.750 ($\pm$9e-4) | 0.846 ($\pm$7e-9) | 0.553 ($\pm$1e-3) | 0.938 ($\pm$7e-7) | 0.915 ($\pm$2e-6) |
| breastw | 0.994 ($\pm$5e-6) | N/A | 0.367 ($\pm$1e-4) | 0.990 ($\pm$9e-6) | 0.995 ($\pm$2e-8) | 0.995 ($\pm$1e-6) | 0.997 ($\pm$2e-7) | 0.994 ($\pm$1e-7) | 0.994 ($\pm$9e-10) | 0.988 ($\pm$2e-5) | 0.995 ($\pm$3e-7) | 0.998 ($\pm$3e-7) |
| campaign | 0.610 ($\pm$3e-4) | 0.603 ($\pm$6e-4) | N/A | 0.745 ($\pm$3e-4) | 0.689 ($\pm$1e-4) | 0.686 ($\pm$1e-3) | 0.689 ($\pm$1e-5) | 0.721 ($\pm$6e-5) | 0.772 ($\pm$4e-8) | 0.710 ($\pm$2e-3) | 0.725 ($\pm$3e-6) | 0.699 ($\pm$2e-6) |
| cardio | 0.979 ($\pm$1e-3) | 0.527 ($\pm$5e-4) | N/A | 0.915 ($\pm$5e-4) | 0.954 ($\pm$5e-6) | 0.913 ($\pm$3e-4) | 0.957 ($\pm$2e-7) | 0.951 ($\pm$9e-4) | 0.946 ($\pm$3e-7) | 0.953 ($\pm$2e-4) | 0.933 ($\pm$8e-6) | 0.977 ($\pm$3e-) |
| census | 0.629 ($\pm$2e-3) | 0.605 ($\pm$2e-4) | N/A | 0.664 ($\pm$2e-4) | 0.605 ($\pm$4e-7) | 0.624 ($\pm$3e-5) | 0.553 ($\pm$2e-5) | 0.611 ($\pm$9e-5) | 0.659 ($\pm$4e-10) | 0.702 ($\pm$2e-4) | 0.661 ($\pm$8e-6) | 0.662 ($\pm$3e-6) |
| fault | 0.928 ($\pm$9e-6) | 0.496 ($\pm$2e-3) | N/A | 0.797 ($\pm$7e-5) | 0.679 ($\pm$4e-5) | 0.716 ($\pm$4e-4) | 0.591 ($\pm$3e-6) | 0.662 ($\pm$1e-4) | 0.485 ($\pm$7e-7) | 0.542 ($\pm$3e-3) | 0.822 ($\pm$4e-6) | 0.884 ($\pm$4e-6) |
| Hepatitis | 0.938 ($\pm$2e-4) | 0.589 ($\pm$6e-3) | 0.625 ($\pm$6e-3) | 0.702 ($\pm$1e-3) | 0.754 ($\pm$3e-3) | 0.555 ($\pm$7e-4) | 0.855 ($\pm$7e-5) | 0.816 ($\pm$7e-4) | 0.786 ($\pm$3e-5) | 0.789 ($\pm$1e-3) | 0.639 ($\pm$7e-4) | 0.855 ($\pm$8e-5) |
| Internetads | 0.878 ($\pm$2e-4) | N/A | N/A | 0.877 ($\pm$2e-4) | 0.689 ($\pm$6e-6) | 0.763 ($\pm$3e-4) | 0.708 ($\pm$6e-7) | 0.425 ($\pm$7e-4) | 0.698 ($\pm$1e-7) | 0.749 ($\pm$1e-3) | 0.823 ($\pm$2e-5) | 0.815 ($\pm$1e-6) |
| landsat | 0.854 ($\pm$8e-4) | 0.580 ($\pm$8e-3) | N/A | 0.819 ($\pm$8e-4) | 0.593 ($\pm$2e-5) | 0.603 ($\pm$8e-3) | 0.471 ($\pm$9e-7) | 0.614 ($\pm$7e-5) | 0.388 ($\pm$3e-7) | 0.462 ($\pm$1e-3) | 0.784 ($\pm$3e-6) | 0.757 ($\pm$2e-6) |
| letter | 0.943 ($\pm$7e-5) | 0.391 ($\pm$6e-4) | N/A | 0.762 ($\pm$2e-3) | 0.757 ($\pm$6e-5) | 0.501 ($\pm$7e-4) | 0.977 ($\pm$5e-6) | 0.639 ($\pm$4e-4) | 0.579 ($\pm$4e-7) | 0.523 ($\pm$2e-3) | 0.917 ($\pm$9e-6) | 0.980 ($\pm$7e-6) |
| mnist | 0.750 ($\pm$3e-3) | 0.615 ($\pm$4e-4) | N/A | 0.974 ($\pm$1e-5) | 0.892 ($\pm$5e-5) | 0.936 ($\pm$2e-5) | 0.789 ($\pm$0) | 0.860 ($\pm$6e-4) | 0.768 ($\pm$6e-7) | 0.834 ($\pm$2e-3) | 0.937 ($\pm$9e-6) | 0.920 ($\pm$2e-5) |
| musk | 1.0 ($\pm$0) | 0.485 ($\pm$2e-2) | N/A | 0.999 ($\pm$2e-8) | 0.999 ($\pm$7e-7) | 0.997 ($\pm$2e-7) | 0.859 ($\pm$0) | 0.960 ($\pm$8e-8) | 0.993 ($\pm$3e-8) | 0.998 ($\pm$5e-6) | 1.0 ($\pm$0) | 1.0 ($\pm$0) |
| shuttle | 0.998 ($\pm$5e-3) | 0.991 ($\pm$2e-3) | N/A | 0.994 ($\pm$1e-5) | 0.992 ($\pm$1e-7) | 0.992 ($\pm$2e-5) | 0.997 ($\pm$6e-9) | 0.996 ($\pm$2e-4) | 0.993 ($\pm$1e-9) | 0.994 ($\pm$6e-6) | 0.995 ($\pm$6e-9) | 0.997 ($\pm$5e-8) |
| speech | 0.859 ($\pm$1e-4) | 0.489 ($\pm$2e-4) | N/A | 0.667 ($\pm$1e-3) | 0.472 ($\pm$7e-7) | 0.486 ($\pm$1e-4) | 0.469 ($\pm$5e-7) | 0.479 ($\pm$2e-4) | 0.473 ($\pm$2e-9) | 0.508 ($\pm$3e-4) | 0.501 ($\pm$1e-5) | 0.881 ($\pm$1e-4) |

Table 19: AP AUC for the different methods on the selected datasets, with variances

| Dataset | CLOE | DAGMM | Og | DRL | RCA | MCM | OC-SVM | iForest | ECOD | DeepSVDD | kNN | KDE |
|---|---|---|---|---|---|---|---|---|---|---|---|
| ALOI | 0.044 (±9e-6) | 0.041 (±2e-5) | N/A | 0.038 (±2e-6) | 0.023 (±4e-4) | 0.042 (±7e-6) | 0.041 (±2e-9) | 0.033 (±9e-8) | 0.032 (±4e-11) | 0.037 (±1e-6) | 0.049 (±7e-6) | 0.042 (±5e-10) |
| backdoor | 0.745 (±2e-2) | 0.033 (±5e-4) | N/A | 0.792 (±1e-3) | 0.102 (±4e-4) | 0.281 (±0.08) | 0.107 (±1e-5) | 0.048 (±6e-5) | 0.093 (±1e-8) | 0.038 (±1e-4) | 0.517 (±1e-3) | 0.411 (±4e-6) |
| breastw | 0.985 (±6e-5) | N/A | 0.204 (±6e-3) | 0.978 (±4e-5) | 0.991 (±9e-8) | 0.991 (±7e-6) | 0.994 (±1e-6) | 0.989 (±6e-7) | 0.987 (±5e-9) | 0.973 (±1e-4) | 0.991 (±1e-6) | 0.996 (±2e-6) |
| campaign | 0.178 (±3e-6) | 0.177 (±2e-4) | N/A | 0.285 (±5e-4) | 0.270 (±1e-4) | 0.266 (±2e-3) | 0.310 (±9e-6) | 0.302 (±1e-4) | 0.356 (±5e-8) | 0.290 (±6e-4) | 0.304 (±9e-6) | 0.296 (±5e-6) |
| cardio | 0.817 (±3e-2) | 0.116 (±4e-3) | N/A | 0.739 (±7e-3) | 0.723 (±2e-4) | 0.587 (±2e-3) | 0.665 (±3e-5) | 0.679 (±2e-3) | 0.626 (±2e-5) | 0.705 (±4e-3) | 0.667 (±8e-5) | 0.861 (±6e-4) |
| census | 0.084 (±9e-5) | 0.086 (±1e-4) | N/A | 0.094 (±1e-4) | 0.070 (±3e-7) | 0.077 (±5e-6) | 0.065 (±3e-7) | 0.074 (±3e-5) | 0.084 (±3e-11) | 0.126 (±2e-3) | 0.084 (±5e-7) | 0.084 (±2e-7) |
| fault | 0.828 (±8e-4) | 0.365 (±1e-3) | N/A | 0.700 (±2e-4) | 0.494 (±7e-6) | 0.588 (±9e-4) | 0.458 (±2e-6) | 0.495 (±8e-5) | 0.337 (±3e-7) | 0.419 (±2e-3) | 0.668 (±9e-5) | 0.825 (±5e-5) |
| Hepatitis | 0.670 (±3e-3) | 0.214 (±6e-3) | 0.361 (±9e-4) | 0.335 (±6e-3) | 0.434 (±0.01) | 0.216 (±2e-4) | 0.395 (±3e-4) | 0.400 (±2e-4) | 0.356 (±9e-5) | 0.439 (±6e-3) | 0.251 (±4e-4) | 0.424 (±2e-5) |
| InternetAds | 0.526 (±6e-3) | N/A | N/A | 0.668 (±4e-3) | 0.501 (±7e-8) | 0.596 (±9e-4) | 0.578 (±2e-6) | 0.155 (±1e-4) | 0.552 (±1e-6) | 0.495 (±4e-3) | 0.692 (±1e-5) | 0.747 (±1e-5) |
| landsat | 0.739 (±4e-4) | 0.267 (±5e-4) | N/A | 0.637 (±4e-3) | 0.246 (±4e-6) | 0.272 (±8e-3) | 0.199 (±2e-7) | 0.273 (±5e-4) | 0.172 (±2e-8) | 0.195 (±2e-4) | 0.473 (±3e-5) | 0.499 (±3e-5) |
| letter | 0.644 (±2e-3) | 0.067 (±1e-5) | N/A | 0.251 (±4e-3) | 0.165 (±1e-4) | 0.078 (±3e-4) | 0.731 (±3e-3) | 0.091 (±5e-6) | 0.079 (±6e-8) | 0.074 (±5e-5) | 0.411 (±2e-4) | 0.723 (±2e-3) |
| mnist | 0.315 (±8e-3) | 0.170 (±6e-4) | N/A | 0.843 (±6e-5) | 0.454 (±1e-4) | 0.735 (±6e-4) | 0.194 (±0) | 0.377 (±2e-3) | 0.194 (±3e-7) | 0.455 (±5e-3) | 0.666 (±5e-5) | 0.640 (±5e-4) |
| musk | 0.999 (±0) | 0.048 (±1e-3) | N/A | 0.990 (±3e-4) | 0.982 (±7e-4) | 0.978 (±2e-3) | 0.104 (±0) | 0.472 (±5e-2) | 0.855 (±1e-5) | 0.941 (±3e-3) | 0.999 (±0) | 0.999 (±0) |
| shuttle | 0.978 (±2e-2) | 0.853 (±2e-3) | N/A | 0.894 (±1e-3) | 0.972 (±4e-6) | 0.841 (±2e-3) | 0.939 (±3e-6) | 0.976 (±1e-5) | 0.912 (±2e-7) | 0.914 (±2e-4) | 0.854 (±7e-6) | 0.875 (±6e-5) |
| speech | 0.068 (±3e-4) | 0.016 (±3e-4) | N/A | 0.044 (±1e-4) | 0.019 (±9e-11) | 0.024 (±6e-5) | 0.019 (±2e-7) | 0.079 (±3e-4) | 0.020 (±4e-10) | 0.017 (±3e-7) | 0.020 (±2e-8) | 0.118 (±7e-4) |

# K LLM USAGE

A LLM was used to check the grammar in the article.

