# OpenReview forum: "CLOE: Christoffel LOss autoEncoder for anomaly detection"
_ICLR.cc/2026/Conference — Submitted to ICLR 2026_

### Official Review · Reviewer_LUtD · 2025-10-19

**Soundness:** 1
**Presentation:** 2
**Contribution:** 2
**Rating:** 2
**Confidence:** 3

**Summary:**

The paper proposes CLOE, a semi-supervised anomaly detection method that learns a low-dimensional latent representation with an autoencoder while regularizing the latent codes with the empirical Christoffel Function (CF). Training proceeds with pretraining (MSE), then joint training with an added CF term, and finally thresholding in latent space using the maximum CF value over the training set. The method targets CPU-friendly operation with a single main hyperparameter $n$, and is evaluated on 10 ADBench tabular datasets using AUROC and AP AUC. On average, CLOE ranks first on AUROC but not on AP AUC, and ablations suggest pretraining is important whereas joint training contributes modestly.

**Strengths:**

- Neat coupling of a CF-based support estimator with AE training via a differentiable loss. Practical, lightweight emphasis (CPU-only).
- Clear method description; principled tie-in to moment matrices and support estimation.
- Algorithms, thresholds, and training stages are spelled out. Datasets and hyperparameters are tabulated.

**Weaknesses:**

- Several influential reconstruction-based methods are missing from the comparisons, including approaches that project along reconstruction pathways, adversarial or probabilistic autoencoder variants, collaborative autoencoders, and recent reappraisals of reconstruction-based OOD detection. Including such baselines would contextualize CLOE within the broader autoencoder AD landscape.
  - Kim, Ki Hyun, et al. "Rapp: Novelty detection with reconstruction along projection pathway." International Conference on Learning Representations. 2019.
  - Almohsen, Ranya, et al. "Generative probabilistic novelty detection with isometric adversarial autoencoders." Proceedings of the IEEE/CVF Conference on Computer Vision and Pattern Recognition. 2022.
  - Pidhorskyi, Stanislav, Ranya Almohsen, and Gianfranco Doretto. "Generative probabilistic novelty detection with adversarial autoencoders." Advances in neural information processing systems 31 (2018).
  - Liu, Boyang, et al. "RCA: A deep collaborative autoencoder approach for anomaly detection." IJCAI: proceedings of the conference. Vol. 2021. 2021.
  - Zhou, Yibo. "Rethinking reconstruction autoencoder-based out-of-distribution detection." Proceedings of the IEEE/CVF Conference on Computer Vision and Pattern Recognition. 2022.
- Recent work has highlighted empirical and theoretical limitations of autoencoder-based anomaly detection and even non-trained baselines. A brief discussion of why CLOE's design might escape these pitfalls would strengthen the framing.
  - Ryu, Seunghyoung, Yonggyun Yu, and Hogeon Seo. "Can untrained neural networks detect anomalies?." IEEE Transactions on Industrial Informatics 20.4 (2024): 6477-6488.
  - Cai, Yu, Hao Chen, and Kwang-Ting Cheng. "Rethinking autoencoders for medical anomaly detection from a theoretical perspective." International Conference on Medical Image Computing and Computer-Assisted Intervention. Cham: Springer Nature Switzerland, 2024.
- **The narrative claims broad superiority, yet the aggregate picture depends on the metric**: CLOE leads the AUROC mean ranking, while DRL leads the AP AUC mean ranking. Without decision-level analyses at the proposed threshold, it is difficult to assert superiority in operational terms. Conclusions should be tempered to reflect metric-dependent outcomes and the absence of threshold-specific evaluation.
- **Threshold policy versus ranking metrics.** The method defines a detection threshold in the latent CF space using a training-set maximum and makes hard inlier–outlier decisions, yet almost all reported evaluation focuses on ranking metrics such as AUROC and AP AUC. This leaves the central thresholding policy empirically unvalidated with respect to operational error profiles. Threshold-dependent metrics such as FPR at fixed TPR, operating points on the precision–recall curve, or F1 at the chosen threshold are needed to substantiate the proposed decision rule. Moreover, setting the threshold as a training maximum risks an overly permissive operating point for the training distribution, with potential overfitting to noise or extreme but normal instances. A validation-quantile or cross-validation–based threshold selection would better reflect deployment.
- **Hyperparameter selection procedure for the polynomial degree.** The rule for choosing the degree relies on a validation loss formed by a reconstruction term plus the mean CF term, accepted if it decreases after an initial burn-in. It is unclear that this surrogate correlates reliably with downstream detection quality, and the acceptance rule is heuristic. The reliance on the mean CF can obscure distributional sensitivity, and the subsequent appendix study shows insensitivity to the Christoffel weight, casting doubt on the practical influence of the CF term. A more systematic selection via grid or Bayesian search over several degrees using validation ROC or PR curves would be more defensible.
- **Fixed latent dimension without justification.** The latent size is fixed to eight for all datasets despite large variation in input dimensionality and complexity. Without a sensitivity study across datasets, it is difficult to assess whether this bottleneck is limiting performance, particularly for very high dimensional data. Reporting results for multiple latent sizes on at least a few datasets would clarify robustness.
- Removing joint training yields only marginal degradation on reported datasets, and the appendix shows the Christoffel weight sweeping across several orders of magnitude does not change outcomes. Together, these results suggest that the CF-guided loss may have limited or null effect in practice, potentially due to scaling, normalization, or gradient-flow issues. Diagnostics such as loss decompositions, gradient norms for the CF term, and controlled ablations across multiple datasets would be needed to confirm effective contribution.

**Questions:**

- Please add comparisons against prominent autoencoder-based AD methods such as projection-path reconstruction, probabilistic adversarial autoencoders, collaborative autoencoders, and recent rethinks of reconstruction-based OOD detection, or justify their exclusion with citations and discussion.
- Briefly explain why CLOE should remain effective under critiques that challenge the reliability of autoencoder-based AD and even untrained baselines, and indicate empirical checks that support this claim.
- Please report confusion-matrix statistics and threshold-dependent metrics at your chosen threshold and at validation-quantile alternatives. How sensitive are deployment metrics to the threshold policy?
- Could you show training curves that separate reconstruction and CF components, together with gradient norms of the CF term? Do latent distributions, margins, or support volumes change when enabling the CF loss compared to a plain autoencoder?
- How well does the validation-loss heuristic predict downstream AP AUC or AUROC across seeds and datasets? Would a small grid over degrees, selected by validation ROC or PR, alter your final configurations?
- Please include results for multiple latent sizes on at least two to three datasets, especially very high-dimensional ones, to rule out capacity bottlenecks.
- Can you align validation-based tuning for baselines under the same protocol and enforce a shared compute and memory budget, reporting success and failure rates accordingly? This would clarify whether the observed gaps persist.

---

> ### Author Response · Authors · 2025-11-21
> **Responses to Weaknesses and Questions, part 1**
>
> We thank the reviewer for its careful reading of our paper and for the constructive and insightful feedback.
>
> **W1** and **A1**
>
> Thank you for the recommendation of these AE-based methods. After reading the different articles, we decided to add RCA [1] to our experimental study. This method is the latest and the closest of our method because it was design for tabular data and not images. For the others methods, Rapp [2] has been developed in 2019, we think it is too old-fashioned to be used in this study. Then GPND [3], GPNDI [4] and [5] have been developed for images. Transformation of a code designed for images to a code designed for tabular data has already been done for the Og method [6] and the results obtained during the tests are very poor, far from the performances obtained by the authors in images. We decided to not introduce other methods designed for image in our study.
>
> [1] Liu, Boyang, et al. "RCA: A deep collaborative autoencoder approach for anomaly detection." IJCAI: proceedings of the conference. Vol. 2021. 2021
>
> [2] Kim, Ki Hyun, et al. "Rapp: Novelty detection with reconstruction along projection pathway." International Conference on Learning Representations. 2019.
>
> [3] Pidhorskyi, Stanislav, Ranya Almohsen, and Gianfranco Doretto. "Generative probabilistic novelty detection with adversarial autoencoders." Advances in neural information processing systems 31 (2018).
>
> [4] Almohsen, Ranya, et al. "Generative probabilistic novelty detection with isometric adversarial autoencoders." Proceedings of the IEEE/CVF Conference on Computer Vision and Pattern Recognition. 2022.
>
> [5] Zhou, Yibo. "Rethinking reconstruction autoencoder-based out-of-distribution detection." Proceedings of the IEEE/CVF Conference on Computer Vision and Pattern Recognition. 2022.
>
> [6] Nicolas Pinon and Carole Lartizien. Ocsvm-guided representation learning for unsupervised anomaly detection. arXiv preprint arXiv:2507.21164, 2025
>
> **W2** and **A2**
>
> In order to answer your question, we will add a study to the ablation study of the revised manuscript with a totally untrained autoencoder used to reduce dimensions and then apply the CF to detect anomalies. Our first tests show that the untrained autoencoder cannot produce a useful representation in its latent space to obtain good performance with the AD CF method.
> For the dataset with a dimension greater than 17, the proposition 1 of [2] is satisfied. For the datasets of dimension 9, we will conduct more experiments to show that the model avoid the identical shortcut.
>
> [2] Cai, Yu, Hao Chen, and Kwang-Ting Cheng. "Rethinking autoencoders for medical anomaly detection from a theoretical perspective." International Conference on Medical Image Computing and Computer-Assisted Intervention. Cham: Springer Nature Switzerland, 2024.
>
> **W3**, **W4** and **A3**
>
> We will add a study of the threshold with the f1 score metric in the revised manuscript. As all the baseline methods consider the contamination rate of the test dataset to find the threshold to compute this metric, we cannot compare our f1-score metric to the other methods.
>
> **A4**
>
> Thank you so much for your feedback. Due to this commentary, we were able to detect a bug in our code. This is now corrected, and the results for all tests will be computed again for the new version of the paper. We will add some training curves in the appendix. Training curves of each loss are decreasing during training. However, the gradient norms of the CF loss are very large, more than 1e7 while the gradient norm of the MSE loss is between 1 and 0. We decided to compute automatically the value of $\lambda$ to have losses of the same order. Thanks to this modification, learning shows better results and performance has improved for high-dimensional datasets. For datasets with a dimension of 9, our conclusions are more mixed and the CF joint training is not required to obtain very good results. We will discuss this in the revised manuscript.

---

> ### Author Response · Authors · 2025-11-21
> **Responses to Weaknesses and Questions, part 2**
>
> **W5** and **A5**
>
> A new version of the paper [7] gives a heuristic to automatically find the parameter $n$ to compute the CF. We have tested this heuristic for all our datasets and the results correspond to the different n considered for the $n$ in the paper. We can add this heuristic to the code to automatically set this parameter. However, training times will increase by a few minutes for large datasets.
>
> [7] Mai Trang Vu, Francois Bachoc, and Edouard Pauwels. Rate of convergence for geometric inference
> based on the empirical christoffel function. ESAIM: Probability and Statistics, 26:171–207, 2022.
>
> **W6** and **A6**
>
> We are limited by the Christoffel function to a maximum size of 8 for the latent space. The computation of the Christoffel function requires the invert of the moment matrices of size $\binom{d+n}{n} \times \binom{d+n}{n}$ (notation $s_d(n)$ in the paper). With $n=6$ and $d=8$, we obtain a size of $3003 \times 3003$ which already takes some times to be computed. That is why we decided to not increase the size of the latent space and we cannot do some tests with a larger latent space on the highest dimensional dataset.
>
> This limitation due to the size of the moment matrices for its inversion is actually in study and will be part of a future work.
>
> **W7** and **A7**
>
> Our method takes more time to train than iForest, ECOD and DRL. We develop this method because it can be trained using only CPU (a study about computation times and cpu memory usage of CLOE will be added to the revised manuscript for each dataset) and it has some theoretical guaranties thanks to the Christoffel function and its empirical version. We will discuss more about the limitations of our method in the revised manuscript. We will also add more datasets in the ablation studies.
>
> Please let us know if you have any further concerns.
>
> Sincerely, Authors

---

### Official Review · Reviewer_JJwm · 2025-10-27

**Soundness:** 2
**Presentation:** 3
**Contribution:** 2
**Rating:** 4
**Confidence:** 4

**Summary:**

This paper proposes a method for semi-supervised anomaly detection combining auto-encoder and Christoffel function. Specially, the method first pretrains an AE using reconstruction loss. Then, the AE is trained by both reconstruction loss and empirical CF-based loss. After training, Christoffel value is used as anomaly score for each test sample.

**Strengths:**

- The paper is well written except some parts of background, making concepts of Christoffel function hard to understand.
- The proposed method is effective in several datasets.
- The concept of Christoffel function is novel in anomaly detection.

**Weaknesses:**

- It seems that the proposed method largely rely on the AE architecture. But what is the motivation for using such architecture instead of a MLP?
- The authors mentioned that the method is lightweight computational yet no theoritical and experimental time cost comparison is provided.
- The authros mentioned that the method is evaluated on 10 high-dimensional datasets yet several datasets have less than 100 and even less than 10 faetures. I do not think such datasets are high-dimensional.
- The baselines are not strong enough. The SOTA methods like [1], [2], [3] and several strong traditional methods like kNN, KDE are not compared.
- Evaluation on more datasets should be included. The authors chose 10 datasets from ADbench for evaluation, yet ADbench includes 47 datasets. Evaluation on more datasets would provide a more comprehensive view of effectiveness of different methods.

[1] Livernoche, Victor, et al. "On diffusion modeling for anomaly detection." arXiv preprint arXiv:2305.18593 (2023).

[2] Yin, Jiaxin, et al. "MCM: Masked cell modeling for anomaly detection in tabular data." The Twelfth International Conference on Learning Representations. 2024.

[3] Thimonier, Hugo, et al. "Beyond individual input for deep anomaly detection on tabular data." arXiv preprint arXiv:2305.15121 (2023).

**Questions:**

- See weaknesses.
- What is the motivation for introducing Christoffel function in AD?
- In Table 3, it seems that the performance of proposed method largely relies on pretraining, what is the possible reason?
- One of the strengths of CF might be that it could figure out a threshold $\gamma_n$ automatically. Yet the authors provide no experiments to evaluate the effectivenss of such threshold compared to the ideal threshold since the two metrics AUROC and AUPRC requires no threshold.

---

> ### Author Response · Authors · 2025-11-21
> **Response to Weaknesses**
>
> We thank the reviewer for the careful reading and its time. Below we address each concern.
>
> **W1**
>
> During the development of our method, we tried to reduce the dimension of the data with PCA and ICA. We also tested using the latent space of a VAE. Performance was poor. We did not try with an MLP because this method has not the reconstruction loss of an autoencoder or of a VAE. This loss encourages the model to learn a meaningful latent space to reconstruct the data and reduce the risk of overfitting the model. Moreover, this loss can improve the performance for our case study, anomaly detection, because the model will not be able to reconstruct abnormal sample.
>
> **W2**
>
> Thank you a lot for this important remark. We will add an array in the appendix of the revised manuscript with the time and the CPU memory usage of the training of CLOE for each datasets. We also put the time for the inference of one sample for each dataset.
> We decided to not measure computational time for other methods because we know that CLOE takes longer than the simpler methods like iForest or ECOD. However, its results are much better for high dimensional dataset, between +0.1 and +0.4 for AU-ROC metric. So by providing the computational times of CLOE with CPU we will prove that our method works on CPU.
>
> **W3**
>
> CLOE is designed to work for tabular data. The benchmark ADBench proposes classical datasets for anomaly detection in tabular data. The larger dimension is more than 1000 and most of the dataset have a dimension lower than 100. To add more high-dimensional datasets, we plan to add in the revised manuscript some results on private data from company of dimensions around 800. CLOE is not designed for images, so we decided not to test with classical dimensions for image applications.
>
> **W4**
>
> Thank you a lot for your suggestions. We will add the reference [2]. [1] looks to be quite different from our method and [3] uses a transformer, thus it cannot work on CPU with large dimensions and answer the industrial constraints we have to meet for our method. Moreover, in order to take into account your suggestions, we added kNN and KDE to the classical methods.
>
> **W5**
>
> We cannot use all the 47 datasets from ADBench because some of them have a dimension of 8 or less. For these datasets, we do not need to reduce the dimension before applying the Christoffel function (CF). We can directly apply the CF on the data and performance should be good according to previous tests performed before the development of the method CLOE. CLOE is not design for this type of data. To answer your request, we will add five more datasets with more than 27 dimensions to our comparison section in the revised manuscript. We will also add the results of the experiments conducted on the company data.
>
> Please let us know if you have any further concerns.
> Sincerely, Authors

---

> ### Author Response · Authors · 2025-11-21
> **Response to Questions**
>
> We thank the reviewer for the careful reading and its time. Below we address each concern.
>
> **A1**
>
> The main motivation stems from the theoretical guarantees brought by the CF. This rational function has a remarkable property: for a fixed value $x$ of its argument, its decrease w.r.t. degree $n$ is at least exponential outside the support of the associated measure and at most polynomial inside the support. This distinguished dichotomy in its behavior explains why its level sets capture the geometric shape of the support, even with low degree. Moreover, for data mining where one usually considers the empirical measure associated with a sample, for fixed degree $n$  and thanks to Strong Law of Large Numbers, the empirical CF has the same properties as the true unknown CF. Of course, when the degree $n$ increases one has to adjust properly the sample size to maintain asymptotics properties of the CF (e.g. related to convergence to density, and support detection) [3].
>
> A previous work [4] considered the CF to detect anomaly in data streams. The results showed better performance than the traditional method KDE for the construction of level sets. These level sets are then used to determine a threshold. The possibility of automatically obtaining a threshold to detect anomalies is a feature that improves the other AD methods tested in this paper.
>
> Another motivation is that the CF only requires one parameter to be tuned, which makes it significantly attractive for real-world applications whole evolution may call for frequent re-learning.
>
> [3] Jean Bernard Lasserre, Edouard Pauwels, and Mihai Putinar. The Christoffel–Darboux kernel for
> data analysis, volume 38. Cambridge University Press, 2022
>
> [4] Kévin Ducharlet, Louise Travée-Massuyès, Jean-Bernard Lasserre, Marie-Véronique Le Lann, and Youssef Miloudi. Leveraging the christoffel function for outlier detection in data streams. International Journal of Data Science and Analytics, pp. 1–17, 2024
>
> **A2**
>
> Thanks a lot for this very interesting remark. After some exploration and new tests on our code, we found an error that can explain this behavior. We correct it and we will redo all the tests for all the datasets and the results will be modified in the revised manuscript.
>
> **A3**
>
> We will add a study on the choice of the threshold with the metric f1-score in the revised manuscript. As all the baseline methods consider the contamination rate of the test dataset to find the threshold to compute this metric, we cannot compare our f1-score metric to the others methods.
>
> Please let us know if you have any further concerns.
>
> Sincerely, Authors

---

> > ### Comment · Reviewer_JJwm · 2025-11-28
> >
> > I thank the authors for their response. However, since no new experimental results have been provided to address my remaining concerns, I maintain my current score.
> >
> > In addition, as noted in **A2**, the code appears to contain errors that may substantially affect the performance of the proposed method and potentially lead to incorrect conclusions. This issue undermines confidence in the experimental findings and the validity of the claims.
> >
> > Given the lack of corrective experiments and the potential impact of code errors on the overall conclusion, I maintain my original score.

---

> > > ### Author Response · Authors · 2025-11-28
> > >
> > > We thank the reviewer for its previous response.
> > >
> > > The code has been corrected and its new version will be added to the revised submission. This revised version will be added in OpenReview next week. The results have been put in replacement of the previous in the revised version. We provide here only the mean for the AP AUC metric due to characters limits:
> > >
> > > AU-ROC for the different methods on the selected datasets
> > > | Dataset     | **CLOE** | **DAGMM** | **Og**  | **DRL** | **RCA** | **MCM** | **OC-SVM** | **iForest** | **ECOD** | **Deep SVDD** | **kNN**   | **KDE**   |
> > > |-------------|----------|-----------|---------|---------|---------|---------|------------|-------------|----------|---------------|-----------|-----------|
> > > | ALOI        | **0.561**| 0.529     | N/A     | 0.523   | 0.546   | 0.534   | 0.517      | 0.539       | 0.531    | 0.546         | _0.556_   | 0.518     |
> > > | backdoor    | **0.944**| 0.619     | N/A     | 0.927   | 0.855   | 0.891   | 0.865      | 0.750       | 0.846    | 0.553         | _0.938_   | 0.915     |
> > > | breastw     | 0.994    | N/A       | 0.367   | 0.990   | 0.995   | 0.995   | _0.997_    | 0.994       | 0.994    | 0.988         | 0.995     | **0.998** |
> > > | campaign    | 0.610    | 0.603     | N/A     | **0.745**| 0.689  | 0.686   | 0.689      | 0.721       | _0.772_  | 0.710         | 0.725     | 0.699     |
> > > | cardio      | **0.979**| 0.527     | N/A     | 0.915   | 0.954   | 0.913   | 0.957      | 0.951       | 0.946    | 0.953         | 0.933     | _0.977_   |
> > > | census      | 0.629    | 0.605     | N/A     | 0.664   | 0.605   | 0.624   | 0.553      | 0.611       | 0.659    | **0.702**     | _0.661_   | 0.662     |
> > > | fault       | **0.928**| 0.496     | N/A     | 0.797   | 0.679   | 0.716   | 0.591      | 0.662       | 0.485    | 0.542         | 0.822     | _0.884_   |
> > > | Hepatitis   | **0.938**| 0.589     | 0.625   | 0.702   | 0.754   | 0.555   | _0.855_    | 0.816       | 0.786    | 0.789         | 0.639     | _0.855_   |
> > > | InternetAds | **0.878**| N/A       | N/A     | _0.877_ | 0.689   | 0.763   | 0.708      | 0.425       | 0.698    | 0.749         | 0.823     | 0.815     |
> > > | landsat     | **0.854**| 0.580     | N/A     | _0.819_ | 0.593   | 0.603   | 0.471      | 0.614       | 0.388    | 0.462         | 0.784     | 0.757     |
> > > | letter      | 0.943    | 0.391     | N/A     | 0.762   | 0.757   | 0.501   | _0.977_    | 0.639       | 0.579    | 0.523         | 0.917     | **0.980** |
> > > | mnist       | 0.750    | 0.615     | N/A     | **0.974**| 0.892  | 0.936   | 0.789      | 0.860       | 0.768    | 0.834         | _0.937_   | 0.920     |
> > > | musk        | **1.0**  | 0.485     | N/A     | _0.999_ | _0.999_ | 0.997   | 0.859      | 0.960       | 0.993    | 0.998         | **1.0**   | **1.0**   |
> > > | shuttle     | **0.998**| 0.991     | N/A     | 0.994   | 0.992   | 0.992   | 0.997      | 0.996       | 0.993    | 0.994         | 0.995     | _0.997_   |
> > > | speech      | _0.859_  | 0.489     | N/A     | 0.667   | 0.472   | 0.486   | 0.469      | 0.479       | 0.473    | 0.508         | 0.501     | **0.881** |
> > > | Mean   | **0.858**| 0.578     | 0.496   | 0.823   | 0.765   | 0.746   | 0.753      | 0.734       | 0.727    | 0.723         | 0.815     | _0.857_   |
> > > | Rank        | 1        | 11        | 12      | 3       | 5       | 6       | 7          | 8           | 9        | 10            | 4         | 2         |
> > >
> > > AP AUC for the different methods on the selected datasets
> > > | Dataset     | **CLOE** | **DAGMM** | **Og**  | **DRL** | **RCA** | **MCM** | **OC-SVM** | **iForest** | **ECOD** | **Deep SVDD** | **kNN**   | **KDE**   |
> > > |-------------|----------|-----------|---------|---------|---------|---------|------------|-------------|----------|---------------|-----------|-----------|
> > > | Mean    | **0.575**    | 0.189     | 0.283   | 0.553   | 0.429   | 0.438   | 0.387      | 0.363       | 0.377    | 0.408         | 0.510     | _0.569_   |
> > > | Rank        | **1**    | 12        | 11      | 3       | 6       | 5       | 8          | 10          | 9        | 7             | 4         | 2         |
> > >
> > > Concerning the issue that the performance of proposed method largely relies on pretraining, the new experiments show that joint training after pretraining improves performance:
> > >
> > > AU-ROC:
> > > | Dataset     | **CLOE** | **Without pretraining** | **Without joint training** | **Untrained AE** |
> > > |-------------|----------|------------------------|----------------------------|------------------|
> > > | Mean    | **0.858**    | 0.802                  | 0.722                      | 0.725            |
> > >
> > > AP AUC:
> > > | Dataset     | **CLOE** | **Without pretraining** | **Without joint training** | **Untrained AE** |
> > > |-------------|----------|------------------------|----------------------------|------------------|
> > > | Mean    | **0.575**    | 0.504                  | 0.369                      | 0.367            |
> > >
> > > Please let us know if you have any further concerns.
> > >
> > > Sincerely, Authors

---

### Official Review · Reviewer_PEKb · 2025-11-01

**Soundness:** 2
**Presentation:** 3
**Contribution:** 3
**Rating:** 4
**Confidence:** 2

**Summary:**

The paper proposes an anomaly detection framework that leverages the **Christoffel Function (CF)** as a regularization term for representation learning. An autoencoder is used for dimensionality reduction, and the training objective combines a reconstruction loss with a CF-based regularization term. The approach aims to achieve effective anomaly detection with fewer hyperparameters compared to prior works.

**Strengths:**

## Pros

* Learning representations guided by the **Christoffel Function** provides a principled way to construct a latent space with certain **theoretical guarantees**, offering a meaningful direction for robust anomaly detection.

**Weaknesses:**

## Cons

* Although the paper claims to use only one hyperparameter (*n*), it actually introduces another regularization coefficient (**λ**) in the training objective, which also serves as a hyperparameter.
* The paper lacks an **ablation study** on the impact of **λ**, which is necessary to assess sensitivity and stability.
* The **evaluated datasets** are limited compared to strong baselines such as **DRL**, which were tested on over 40 datasets. It is also unclear whether the baselines were trained with the **same number of samples** for fair comparison.
* The **limitations** of the proposed method are not discussed, including potential issues in scalability, representation quality, and robustness in high-dimensional or real-world data.
* The paper lacks important **experimental details**, such as the architecture of the autoencoder and the variance of reported performance metrics in the results table.

**Questions:**

See Weaknesses section.

---

> ### Author Response · Authors · 2025-11-21
> **Response to Weaknesses**
>
> We thank the reviewer for its positive and constructive feedback. Below we provide detailed responses to its concerns.
>
>  **W1** and **W2**
>
>  $\lambda$ is a hyperparameter. However, the tests in appendix D show that its value does not influence too much the model. That is why we decided to not consider it as a hyperparameter. A slight modification of our loss combination leads us to remove the choice of lambda and to compute it automatically with the gradient norm of our two losses because the gradient norm of the Christoffel function (CF) loss has values much higher than the gradient norm of the MSE loss, in the order of 1e7 for the gradient norm of the CF and in the order of 1 for the gradient norm of the MSE loss. The new results will be presented in the revised manuscript and lambda will no longer be a hyperparameter.
>
>
> **W3**
>
> We decided to use datasets for evaluation that have a dimension greater than 9. Our method is for high dimensional datasets ($>9$): to use the CF on low dimensional datasets, we do not need to reduce the data dimension, we can directly apply the CF on the real data without any dimension reduction step. On these low dimensional datasets, CF shows very good results. That is why we do not consider all the datasets that DRL have tested. We will add 5 more high dimensional datasets from ADBench to our tests and the result of an experiment conducted on private company data on the revised manuscript.
>
> **W4**
>
> We will add a paragraph about the limitation of the method in the revised manuscript.
> Concerning scalability, the condition of the size of the latent space is needed to respect the condition to inverse the moment matrix to compute the CF. This condition can be limiting for very high-dimensional datasets. However, our tests obtain good results even on real datasets and we decide to keep this restriction about the latent space.
>
> **W5**
>
> The architecture of the autoencoder is described in section 5.4, l.348. Could you please precise what other information about the architecture you want us to add? We will be glad to add information.
>
> We will add the details about the variance of the reported metrics in the appendix of the revised manuscript.
>
> Please let us know if you have any further concerns.
> Sincerely, Authors

---

### Comment · Area_Chair_FHzj · 2025-11-28

Dear Reviewers,

Thank you for your valuable time and expertise in reviewing this paper.

The authors have now submitted their rebuttal. We would appreciate it if you could review their responses and assess whether your concerns have been addressed.

Best regards,

AC

---

### Author Response · Authors · 2025-12-03
**General Rebuttal Comment**

Dear Area Chair and Reviewers,

We thank the reviewers and Area Chairs for the time and effort they dedicated to the evaluation of our paper and for providing the final assessment. They have greatly helped us in improving the presentation and results of our work.

A summary of the revisions made to the paper in relation with each of the reviewers comments is provided below. All the modifications in the revised paper are marked in blue to ease proofreading.

Summary of Key Revisions and New Results in the revised PDF uploaded:

- As recommended by reviewers JJwm and LUtD, 4 new methods kNN, KDE, RCA [1] and MCM [2] have been added in the related work and experiment sections as baseline methods. CLOE maintains the best performances, after KDE.

- As recommended by PeKb and JJwm reviewers, five more datasets from ADBench have been added in the experiments: *ALOI*, _mnist_, _musk_, _campaign_, and _census_. These datasets have been chosen according to their dimensions, 27, 100, 166, 62, and 500, respectively, because they meet the specific requirements of CLOE, in particular the fact that CLOE is designed for datasets high dimension datasets). CLOE obtains good results for these datasets except for \textit{mnist}, a dataset derived from the image MNIST dataset. Lower performance may be explained by the fact that CLOE is not designed to detect anomalies in images.

- All experiments have been replayed and the new results are displayed in Tables 1 and 2. Details about the variance are presented in Appendix J, Table 18 and 19. CLOE obtains the best performance for both metrics in mean. Figure 2 in Appendix H presents the monitoring of the training curves for the two loss components through the epochs, as asked by reviewer LUtd. These curves decrease, which confirms the impact of the MSE loss and the CF loss during CLOE training.

- The ablation studies have been recomputed on all the datasets and a new study has been added. The new study considers the untrained AE to encode data. The results show that CLOE with untrained AE has lower performance than CLOE with trained AE, which is not in alignment with paper [3] indicated by reviewer LUtD. This may originate from the fact that [3] deals with an MLP and not and AE.

- As highlighted by reviewer PEKb, the coefficient $\lambda$ weighting the regularization term in the loss could be considered as a hyperparameter in the first version of the paper. In the revised version, we propose a method to automatically obtain $\lambda$ at each epoch from the norm of the MSE loss gradient and the CF loss gradient when it is non-zero. $\lambda$ is thus no longer a hyperparameter, which means that CLOE only has one hyperparameter "n".

- As recommended by reviewers JJwm and LUtD, an analysis about the anomaly threshold based on the F1-score has been added to confirm the relevance of the automatic threshold selection of CLOE. The results show that the threshold selected by CLOE outperforms most thresholds proposed in the existing literature.

- To prove that CLOE is a lightweight method, training time, CPU memory usage, and testing time—both for the full dataset and for a single sample—are reported in Appendix C, Table 7 for all datasets.

- The hyperparameters of the baseline methods  are detailed in Appendix E, in Table 9, as recommended by reviewer PEKb.

- A new experiment with real datasets has been added in Appendix I as recommended by reviewer PEKb.

We hope these revisions improve the coherence, clarity, and transparency of the concepts and method conveyed.

Best regards,

Authors of Paper Submission 12957

[1] Liu, Boyang, et al. "RCA: A deep collaborative autoencoder approach for anomaly detection." IJCAI: proceedings of the conference. Vol. 2021. 2021.

[2] Yin, Jiaxin, et al. "MCM: Masked cell modeling for anomaly detection in tabular data." The Twelfth International Conference on Learning Representations. 2024.

[3] Ryu, Seunghyoung, Yonggyun Yu, and Hogeon Seo. "Can untrained neural networks detect anomalies?." IEEE Transactions on Industrial Informatics 20.4 (2024): 6477-6488.

---

### Meta-Review · Area_Chair_9WfZ · 2026-01-06

**Summary:**

The reviewers raise several concerns regarding clarity, experimental rigor, and substantiation of claims. First of all despite stating that the method uses only one hyperparameter, an additional regularization coefficient (λ) is introduced without ablation or sensitivity analysis, and key design choices, i.e., the autoencoder architecture, fixed latent dimensionality, and polynomial degree selection, lack justification or systematic validation. The evaluation is limited in scope, using only a small subset of ADbench datasets (many of which are not truly high-dimensional) and comparatively weak baselines, while omitting several strong classical, deep, and recent reconstruction-based anomaly detection methods, as well as discussion of known limitations of autoencoder-based approaches. Reported performance advantages are metric-dependent and rely primarily on ranking metrics (AUROC, AP AUC), even though the method defines an explicit decision threshold, leaving the proposed thresholding policy empirically unvalidated and potentially prone to overfitting. Moreover, ablations suggest that key components, including the Christoffel-function–guided loss, may have limited practical impact, calling into question the method’s core contribution and warranting deeper diagnostic analyses, broader benchmarking, and more cautious conclusions.

**Reviewer Concerns:**

The authors have provided a substantial rebuttal in the attempt to address most of the concerns from the reviewers. I agree that the answers are good but still there are several things not really clarified. True that there was no real discussion with the reviewers (desired from the very beginning) but for example Reviewer JJwm already indicated that some of the issues were still there. The main issues acknowledged also in the rebuttal were the novel contribution and the limited evaluation. The authors have provided some substantial material to address these pointes but I feel like this should have already been in the original manuscript given their importance and amount.

**Reviewer Scores:**

I think the discussion might have improved some of the opinions of the reviewers but I doubt the overall outcome would have been different. The main problem is that all the initial scores were rather negative and the concerns were real. This has been acknowledged also by the authors in their rebuttal. Overall, I think the authors have done a good job in their rebuttal but the starting point was too low to change the final decision.

---

### Decision · Program_Chairs · 2026-01-26

Reject